# DIFFERENTIALLY PRIVATE FEDERATED CLUSTERING WITH RANDOM REBALANCING

## ABSTRACT

Federated clustering aims to group similar clients into clusters and produce one model for each cluster. Such a personalization approach typically improves model performance compared with training a single model to serve all clients, but can be more vulnerable to privacy leakage. Directly applying client-level differentially private (DP) mechanisms to federated clustering could degrade the utilities significantly. We identify that such deficiencies are mainly due to the difficulties of averaging privacy noise within each cluster (following standard privacy mechanisms), as the number of clients assigned to the same clusters is uncontrolled. To this end, we propose a simple and effective technique, named RR-Cluster, that can be viewed as a light-weight add-on to many federated clustering algorithms. RR-Cluster achieves reduced privacy noise via *randomly rebalancing* cluster assignments, guaranteeing a minimum number of clients assigned to each cluster. We analyze the tradeoffs between decreased privacy noise variance and potentially increased bias from incorrect assignments and provide convergence bounds for RR-Cluster. Empirically, we demonstrate that RR-Cluster plugged into strong federated clustering algorithms results in significantly improved privacy/utility tradeoffs across both synthetic and real-world datasets.

## 1 INTRODUCTION

Clustering is a critical approach for adapting to varying client distributions in federated learning (FL) by learning one model for each group of similar clients (Li et al., 2020a; Kairouz et al., 2021). It usually outperforms the canonical FL formulation of learning a single global model, as it outputs several models to serve heterogeneous client population (Ghosh et al., 2020). Federated clustering typically works by identifying clusters clients belong to and incorporating model updates into the corresponding cluster models iteratively. However, performing clustering requires clients to transmit additional data-dependent information, making the system more vulnerable to privacy leakage.

In this work, we aim to finally output $k$ cluster models such that the set of these $k$ models satisfies global client-level differential privacy (DP) (Dwork et al., 2006; McMahan et al., 2018), with a trusted central server. Despite recent attempts privatizing federated clustering by DP (Li et al., 2023; Augello et al., 2023), we identify that a fundamental tension between DP and clustering still remains—DP tries to hide individual client's contributions to the output models, whereas clustering may expose more client information (i.e., client model updates), especially if the cluster size is small (or even one). We need much larger effective noise to privatize those clients that have a huge impact on their assigned clusters, which can degrade the overall privacy/utility tradeoffs. For instance, we observe significant utility drops if directly applying standard DP mechanisms on top of existing federated clustering algorithms (Figure 1 and Section 5).

Inspired by these insights, we propose a simple and light-weight technique, RR-Cluster, that avoids too small clusters by randomly sampling a subset of client model updates belonging to large clusters into small ones, thus guaranteeing each cluster having at least $B$ assigned model updates at each round. Hence, the effective privacy noise after averaging the updates would become smaller. Such data-independent random rebalancing step can be applied on top of various clustering algorithms where the server aggregates model updates within each cluster based on learnt cluster assignments, without extra privacy or communication costs. We note that a key hyperparameter here is $B$, the lower bound of the number of clients assigned to each cluster. When $B=0$, it recovers any base clustering algorithm. When $B$ increases, we reduce privacy noise at the cost of potentially increasing

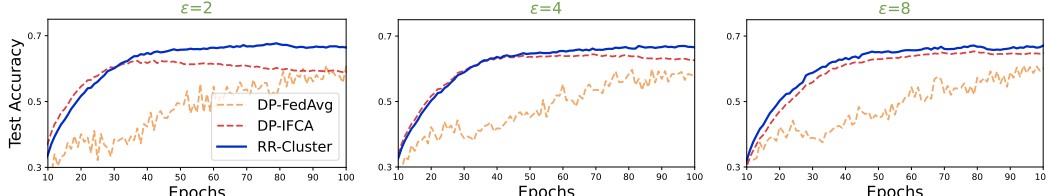

Figure 1: Test accuracy on FashionMNIST (Xiao et al., 2017) with a neural network model. Directly privatizing previous clustering method (Ghosh et al., 2020) (DP-IFCA) degrades model performance significantly, even underperforming the non-personalized approach (DP-FedAvg (McMahan et al., 2018)) for some privacy budgets. The proposed `RR-Cluster` plugged into IFCA achieves higher accuracy in private settings under various $\varepsilon$'s.

clustering bias by moving clients from correct clusters to incorrect ones. Theoretically, we analyze the effects of $B$ in the convergence bounds when plugging `RR-Cluster` into a competitive federated clustering algorithm (Ghosh et al., 2020). Empirically, we find that (1) small values of $B$ can improve privacy/utility tradeoffs significantly, (2) and that the potential clustering bias from `RR-Cluster` does not hurt the learning process too much, especially in cases where the original assignment before rebalancing may not be accurate.

Our contributions are summarized as follows. We propose `RR-Cluster`, a simple technique to improve privacy/utility tradeoffs for federated clustering by randomly sampling model updates from large clusters to small ones (Section 3). It can be used as a light-weight add-on on top of various federated clustering algorithms. We theoretically analyze the privacy and convergence bounds of `RR-Cluster` in convex settings, revealing a tradeoff between reduced privacy noise and increased bias (Section 4). Empirically, we demonstrate the effectiveness of `RR-Cluster` across both real-world and synthetic federated datasets under natural and controlled heterogeneous data distributions (Section 5). `RR-Cluster` outperforms vanilla private baselines by a large margin.

## 2 RELATED WORKS AND BACKGROUND

**Federated Clustering and Personalization.** Federated clustering is a critical approach for federated networks by learning one model for each group of similar clients, which can benefit various downstream tasks including personalization (Dennis et al., 2021). The IFCA method (Ghosh et al., 2020) is one of the most popular clustering algorithm, where each client 1) computes the losses of all cluster models on the local dataset and 2) selects the cluster having model of minimum test loss as their belonged cluster. Other clustering approaches leverage different distance measures for the cluster assignment (Long et al., 2023) or incorporate an additional global model to improve performance (Ma et al., 2023). While these clustering methods significantly enhance the performance of federated systems, challenges remain to guarantee differential privacy (DP) for the entire learning pipeline. In Section 5, we show how plugging `RR-Cluster` into some of these clustering methods can achieve better test performance relative to directly privatizing these clustering methods.

**Differentially Private Distributed/Federated Clustering.** It is critical to ensure (differential) privacy when clustering distributed and sensitive data, which has been studied extensively in prior works mostly for data analytics (e.g., Nissim and Stemmer, 2018; Stemmer, 2021; Huang and Liu, 2018; Balcan et al., 2017; Chang et al., 2021; Cohen-Addad et al., 2022). However, joint clustering and *learning* algorithms under privacy constraints are generally less explored. In the context of federated learning, existing works have considered private federated clustering in different settings. For instance, the DCFL algorithm (Augello et al., 2023) combines a dynamic clustering method with DP under global DP. Malekmohammadi et al. (2025) propose DPCFL, which uses full-batch local training in the first round and a GMM-based soft clustering of client updates to mitigate the impact of DP noise. However, DPCFL focuses on the *sample-level DP* setting, whereas our work targets *client-level global DP* and addresses a different set of privacy and clustering challenges. We note that some works may provide insufficient privacy protection without privatizing all the raw-data-dependent intermediate information in the clustering process (e.g., cluster assignment identifiers) (Li et al., 2023; Luo et al., 2024). Despite the existence of private federated clustering algorithms, issues related to high sensitivity of averaged model updates that belong to small clusters (thus resulting in large privacy noise) still remain. Some works target at sample-level DP (Liu et al., 2022) or local DP (He et al., 2023), different from our setting focusing on client-level global DP with a trusted central server, as defined below.

**Definition 1** (Differential Privacy (DP) (Dwork et al., 2006)). *A randomized algorithm $\mathcal{M} : U \to \mathbb{R}^{k \times d}$ is $(\varepsilon, \delta)$-DP if for all adjacent datasets $D, D' \in U$ and the algorithm output $O \in \mathbb{R}^{k \times d}$,*

$$\Pr(\mathcal{M}(D) \in O) \le e^{\varepsilon} \cdot \Pr(\mathcal{M}(D') \in O) + \delta.$$

For client-level global DP, we define adjacent dataset $D'$ and $D$ by adding or removing any single client, which is a commonly-used privacy notion in federated settings (McMahan et al., 2018). In the clustering setting, we consider the output $\mathcal{M}(D) \in \mathbb{R}^{k \times d}$ to be $k$ clustered models, each in $\mathbb{R}^d$.

Additionally, we mainly adopt Rényi differential privacy (RDP) (Mironov, 2017) for privacy accounting. For completeness, we include its formal definition below, and restate the existing RDP-to-DP conversion theorem in Theorem 3 in the Appendix.

**Definition 2.** *($(\lambda, \epsilon)$-Rényi Differential Privacy (Mironov, 2017)) A randomized mechanism $\mathcal{M} : U \to \mathbb{R}^{k \times d}$ is $(\lambda, \epsilon)$-RDP if for all adjacent datasets $D, D' \in U$ and the algorithm output $O \in \mathbb{R}^{k \times d}$,*

$$D_{\lambda}(\mathcal{M}(D) || \mathcal{M}(D')) = \frac{1}{\lambda - 1} \log \mathbb{E}_{o \sim \mathcal{M}(D)} \left[ \left( \frac{\Pr(\mathcal{M}(D) = o)}{\Pr(\mathcal{M}(D') = o)} \right)^{\lambda - 1} \right] \le \epsilon.$$

**Threat Model and Privacy Setting.** In this work, we follow the standard and widely-adopted setting of client-level differential privacy with a trusted central server (McMahan et al., 2018; Kairouz et al., 2021). Our goal is to protect client data from a external adversary, who can observe the final output of the entire federated learning process, which in our case is the set of $k$ final cluster models, $\{\theta_j^T\}_{j \in [k]}$. We note that this setting is different from local DP, where the server is untrusted and clients privatize their updates before transmitting information. In our threat model, the central server is trusted and is not considered an adversary. We treat an external observer as the adversary, who only sees the final cluster models and attempts to infer whether any particular client participated in the federated training.

## 3 PRIVATE FEDERATED CLUSTERING WITH RANDOM REBALANCING

In this section, we introduce the general framework of `RR-Cluster` (Section 3.2) and present an instantiation of `RR-Cluster` by plugging it into a popular clustering method IFCA (Ghosh et al., 2020) (Section 3.3). Finally, we discuss tradeoffs introduced by `RR-Cluster` around privacy noise reduction and clustering bias (Section 3.4).

### 3.1 PROBLEM FORMULATION

In this work, we study the common federated clustering objective, i.e., grouping $M$ clients into $k$ groups and outputting one model for each group. We assume there are $k$ underlying data distributions $\{\mathcal{D}^0, \cdots, \mathcal{D}^{k-1}\}$, and the $M$ client indices are partitioned into $k$ disjoint sets $\{S_0, \cdots, S_{k-1}\}$ indicating which clusters (distributions) clients belong to. Local data $D_i$ of client $i \in [M]$ follow the distribution $\mathcal{D}^j$ if $i \in S_j$. Let $f(\theta; z)$ be the loss function with model parameter $\theta \in \mathbb{R}^d$ and data point $z$. For each client $i \in [M]$, we define their local empirical loss $F_i(\theta)$ as $F_i(\theta) := \frac{1}{|D_i|} \sum_{z \in D_i} f(\theta; z)$. The optimization objective of cluster $j$ is: $F^j(\theta) := \frac{1}{|S_j|} \sum_{i \in S_j} F_i(\theta), j \in [k]$. We search for optimal parameters $\theta_j^*$ for each cluster $j \in [k]$ as $\theta_j^* := \arg\min_\theta F^j(\theta)$. For federated clustering algorithms, the server typically maintains cluster models $\{\theta_j\}_{j \in [k]}$, which are periodically aggregated from corresponding client models based $\{S_j\}_{j \in [k]}$.

### 3.2 RR-CLUSTER ALGORITHM

We introduce the differentially private federated clustering framework and the proposed `RR-Cluster` method. As `RR-Cluster` is compatible with many clustering methods, we present a general algorithm to illustrate the main idea in this section and present an instantiation with IFCA (Ghosh et al., 2020) in Section 3.3.

At a high level, federated clustering jointly learns $k$ cluster model parameters and updates cluster assignments of clients, as summarized in Algorithm 1. In the $t$-th round, server samples $qM$ clients, and sends the current $k$ cluster models to each of them. Each client $i$ identifies the cluster they belong to and optimizes that cluster model's parameters using local data. They then send the model updates $\Delta \theta_i^t$ to the server as well as other information necessary for clustering (Line 5). Without our proposed plug, the server would identify model updates corresponding to each cluster (Line 7), privatize and

---

**Algorithm 1:** The Proposed Method: `RR-Cluster`

---

1  **Input:** total communication round $T$, number of clusters $k$, number of clients $M$, init clusters
    model $\{\theta_j^0\}_{j\in[k]}$, client sampling rate $q$, parameter clipping bound $C_\theta$, rebalancing
    threshold $B$ ($1 \le B \le \frac{qM}{k}$). The specific $s_i$ mentioned in Line 5 depends on the
    underlying clustering algorithm. See details of $s_i$ in Algorithm 2.

2  **for** $t = 0, \cdots, T-1$ **do**

3  $\quad$ Server samples a subset of clients $M^t$ with probability $q$

4  $\quad$ Server sends $\{\theta_j^t\}_{j\in[k]}$ to sampled clients

5  $\quad$ Client $i \in M^t$ sends back model updates $\Delta\theta_i^t$ and other data-dependent information $s_i$

6  $\quad$ Server privatizes $\{s_i\}_{i\in M^t}$ into $\{\widetilde{s}_i^t\}_{i\in M^t}$ if $\{s_i\}_{i\in M^t}$ directly queries raw client data

7  $\quad$ Server partitions $M^t$ into $k$ groups (with client indices in $\{S_j^t\}_{j\in[k]}$) based on $\widetilde{s}_i$

8  $\quad$ Server determines large/small groups comparing $|S_j^t|$ with $B$, $j \in [k]$

9  $\quad$ Server samples clients from large groups to insert into small ones, i.e., rebalancing cluster
    $\quad$ assignments s.t. $|S_j^t| \ge B$ for all $j \in [k]$

10 $\quad$ Server clips client updates $\Delta\bar{\theta}_i^t \leftarrow \frac{\Delta\theta_i^t}{\max(1, \|s\|/C_\theta)}$, $i \in M^t$

11 $\quad$ Server aggregates client updates $\Delta\widetilde{\theta}_j^t \leftarrow \frac{1}{|S_j^t|}\left(\sum_{i\in S_j^t} \Delta\bar{\theta}_i^t + \mathcal{N}(0, (2C_\theta)^2\sigma_\theta^2)\right)$, $j \in [k]$

12 $\quad$ Server uses noisy updates to update cluster models $\theta_j^{t+1} \leftarrow \theta_j^t + \gamma\Delta\widetilde{\theta}_j^t$, $j \in [k]$

13 **return** *cluster models* $\{\theta_j^T\}_{j\in[k]}$

---

aggregate those updates within each cluster (Line 10 to Line 12), and output new cluster models $\{\theta_j^T\}_{j\in[k]}$. This procedure directly privatizes existing federated clustering algorithms.

However, one fundamental challenge associated with these clustering algorithms is that the cardinality of the set of model updates corresponding to each cluster (i.e., $|S_j^t|, j \in [k]$) is not controlled. Therefore, the privacy noise needed to hide client model updates within each cluster (i.e., effective noise added after averaging model updates) could be large. To address this issue, we propose a simple and effective plug-in (highlighted in red) that uniformly randomly samples model updates from large groups (i.e., $|S_j^t| > B$) and move them to small ones. With client sample rate $q$, we guarantee that each cluster has a minimum of $B$ client model updates ($1 \le B \le qM/k$), reducing privacy noise by averaging across at least $B$ updates to update $\theta_j^t$. The resulting private clustering framework is named `RR-Cluster`. Note that we can set $B$ so that any client update can only be sampled and reassigned to another cluster at most once. Note that given the modularity of `RR-Cluster`, it can preserve the communication-efficiency of any federated clustering method that it is plugged into. Furthermore, we discuss the tradeoffs introduced by the rabanlancing threshold $B$ in Section 3.4.

### 3.3 RR-CLUSTER PLUGGED INTO IFCA

Having established the high-level structure of `RR-Cluster`, we now plug the random rebalancing idea into one popular clustering method IFCA (Ghosh et al., 2020) as an example, presented in Algorithm 2. We also empirically evaluate the performance of `RR-Cluster` plugged into other federated clustering algorithms in Section 5. In IFCA, at each iteration, each client independently determines the cluster they belong to by selecting the model with the smallest loss evaluated using local data, as shown in Line 5. Each selected client $i$ additionally communicates their cluster identifier to the server organized as a one-hot vector $s_i \in \mathbb{R}^k$, which is privatized along with model parameters.

**Other Differentially Private Mechanisms.** There are various privatization algorithms for model parameters $\theta$ and cluster identifier $s_i$'s. `RR-Cluster` is agnostic of the specific choice of private mechanisms. For $\theta \in \mathbb{R}^d$, it is a common practice to use the Gaussian mechanism after clipping to upper bound the $L_2$ norm of parameters. However, for cluster identifiers $s_i^t$, which is a $k$-dimensional one-hot discrete vector, it is also reasonable to adopt the exponential mechanism. Though we use the Gaussian mechanism in experiments (Section 5), `RR-Cluster` achieves better performance regardless of how to privatize identifiers, as it reduces DP noise for model parameters.

### 3.4 EFFECTS OF $B$: TRADEOFFS BETWEEN PRIVACY NOISE AND CLUSTERING BIAS

Our method mainly benefits from the reduction of privacy noise after inserting model updates sampled from large groups $\{S_j\}_{j\in j_l}$ into the small ones $\{S_j\}_{j\in j_s}$ (Line 11). However, this approach may

---

**Algorithm 2:** `RR-Cluster` (IFCA)

1    **Input:** The same as Algorithm 1

2    **for** $t = 0, \cdots, T-1$ **do**

3       Same steps as in Algorithm 1, Line 3 - Line 4

4       **for** *client* $i \in M^t$ *in parallel* **do**

5          Determines its cluster: $\hat{j} \leftarrow \arg\min_{j \in [k]} F(\theta_j^t, D_i)$ and embeds $\hat{j}$ into one-hot: $s_i^t \leftarrow \mathbf{1}_{\{j=\hat{j}\}}$

6          $\theta_i^t \leftarrow \text{Local Training}(D_i, \theta_{\hat{j}}^t)$,   $\Delta\theta_i^t \leftarrow \theta_i^t - \hat{\theta}_{\hat{j}}^t$

7          Send $\Delta\theta_i^t$ and $s_i^t$ back to server

8       Server privatizes $s_i$ as $\widetilde{s}_i^t \leftarrow \frac{s_i^t}{\max(1, \|s\|/C_s)} + \mathcal{N}(0, C_s^2 \sigma_s^2)$ for $i \in M^t$

9       Server updates cluster assignments: $S_j^t.\texttt{append}(i)$ where $j \leftarrow \arg\max_{j \in [k]} \widetilde{s}_i[j]$ for $i \in M^t$

10      Server determines large/small groups $j_l = \{j \mid |S_j^t| \geq B, j \in [k]\}$; $j_s = [k] \setminus j_l$

11      Server samples client indices assigned to large clusters $\bigcup_{j \in j_l}\{S_j^t\}$ uniformly at random to expand small ones $\{S_j^t\}_{j \in j_s}$ such that $|S_j^t| = B$ for $j \in j_s$

12      Server privatizes $\Delta\theta_i^t$ and update cluster model $\theta_j^t$ as in Algorithm 1, Line 10 - Line 12

13   **return** *cluster models* $\{\theta_j^T\}_{j \in [k]}$

---

introduce bias if the updates that are initially correctly assigned to large groups are incorrectly re-assigned to one of the small ones. We discussed this issue both theoretically (Section 4.2) and empirically through a case study on synthetic data (Appendix E.5). Empirically, we observe that benefits of reduced privacy noise outweigh potential bias.

Additionally, we note that the simple random rebalancing step may not introduce much error or bias, as the initial cluster assignment (before rebalancing) may be incorrect anyway, which is a common issue for (federated) clustering sometimes known as model collapse (Wu et al., 2022). That is, the clustering algorithm starts with $k$ models but only a subset of them get effectively trained and the solutions get gradually biased towards those useful models. As empirically demonstrated in Table 11, a subset of models can collapse after training. As our method maintains a minimum number of updates assigned to each cluster, one side effect is that it ensures each cluster receives some updates and thus would not be under-trained even if the updates may be slightly biased. Though `RR-Cluster` is designed for private training, in Section 5, we show that our method remains competitive or even outperforms other baselines in non-private training as well, due to this side effect.

## 4   THEORETICAL ANALYSIS

### 4.1   PRIVACY ANALYSIS

We state the privacy guarantees for the proposed method Algorithm 1 in terms of user-level global DP and RDP. In each communication round, conditioned on a randomly sampled subset of clients, there are two other random processes: (1) privatizing cluster identifiers and (2) privatizing model parameters. Hence, the privacy loss per round is accumulated from the composition of these two processes and the client subsampling step. First, we have the following proposition.

**Proposition 1.** *At each round of* `RR-Cluster`*, given a subset of selected clients* $M^t$ *as input, a randomized mechanism* $\mathcal{M}$ *that outputs* $k$ *new clustered models satisfies* $(\alpha, \varepsilon_1 + \varepsilon_2)$*-RDP if: (1) we set the noise scale for model parameters to* $\sigma_\theta = \sqrt{\frac{\alpha}{2\varepsilon_2}}$ *and (2) we use any cluster identifier privatization method that satisfies* $(\alpha, \varepsilon_1)$*-RDP.*

The above result directly follows from adaptive composition properties of RDP (Mironov, 2017) combined with sensitivity bound of the sum of model updates within each cluster (Appendix B.2). As discussed in Section 3.2, we can adopt existing randomized algorithms to privatize cluster identifiers, as long as it achieves $(\alpha, \varepsilon_1)$-RDP. Using these noisy cluster identifiers, we identify large and small clusters and perform random rebalancing. Given rebalanced clusters (i.e., disjoint partitions of selected clients), we privatize each cluster model with a Gaussian mechanism to obtain $(\alpha, \varepsilon_2)$-RDP for each model. Since these $(\alpha, \varepsilon_2)$-RDP models are trained on disjoint client partition in parallel, we achieve overall $(\alpha, \varepsilon_2)$-RDP for the output of $k$ models per round conditioned on subsampled clients.

To derive the final privacy guarantee of RR-Cluster, we analyze (1) the client sampling process and (2) the composition of $T$ communication rounds. Note that the randomized mechanism $\mathcal{M}$ in Proposition 1 takes as input the subset of clients $M^t$. Thus, given the input of total $M$ clients, we can exploit randomness during client selection and derive a stronger privacy guarantee. Consider $\mathcal{M}$ as a black-box mechanism, we can apply privacy amplification theorem of RDP under sampling without replacement (Wang et al., 2019). We then apply composition theorem (Mironov, 2017) to account for total $T$ communication rounds and derive the final RDP privacy guarantee as follows.

**Theorem 1.** *Suppose a randomized mechanism $\mathcal{M}$ that inputs a subset of selected clients $M^t$ and outputs $k$ new clustered models satisfies $(\alpha, \varepsilon_1(\alpha)+\varepsilon_2(\alpha))$-RDP for a single communication round of RR-Cluster. Given a client sampling ratio $q$ and total $T$ communication rounds, RR-Cluster algorithm satisfies $(\alpha, \varepsilon(\alpha))$-RDP where*

$$\varepsilon(\alpha) \leq \frac{T}{\alpha-1} \cdot \log\left(1 + q^2\binom{\alpha}{2}\min\left\{4(e^{\varepsilon_1(2)+\varepsilon_2(2)}-1), \quad e^{\varepsilon_1(2)+\varepsilon_2(2)}\min\{2,(e^{\varepsilon_1(\infty)+\varepsilon_2(\infty)}-1)^2\}\right\}\right.$$

$$\left. + \sum_{j=3}^{\alpha} q^j\binom{\alpha}{j}e^{(j-1)(\varepsilon_1(j)+\varepsilon_2(j))}\min\{2,(e^{\varepsilon_1(\infty)+\varepsilon_2(\infty)}-1)^j\}\right).$$

Here, $\varepsilon$ is denoted as $\varepsilon(\alpha)$, as $\varepsilon$ is functionally determined by the choice of $\alpha$. We note that the above complex bound directly follows from Wang et al. (2019), which provides a precise non-asymptotic bound rather than relying on an asymptotic bound. In Appendix B.3, we provide a detailed proof of Theorem 1 and the privacy guarantee in terms of DP using RDP-DP conversion (Mironov, 2017).

### 4.2 CONVERGENCE OF RR-CLUSTER

In this section, we provide convergence guarantees for our proposed Algorithm 2 (RR-Cluster plugged into IFCA (Ghosh et al., 2020)) and theoretically explore the tradeoffs between increased clustering bias and reduced privacy noise as a result of random rebalancing with a parameter $B$.

For simplicity, our convergence analysis assumes all clients participate in every round. Our theoretical guarantees rely on several standard assumptions (as detailed in Appendix C). We assume the loss functions are $L$-smooth and $\lambda$-strongly convex. We also assume the variance of loss functions, variance of gradients, and variance of cluster sizes are bounded by $\eta^2$, and $v^2$, and $\mu^2$, separately. For the differential privacy analysis, the clipping threshold $C_\theta$ is assumed to be larger than norm of the natural gradient, meaning that the clipping operation does not affect the model update. Furthermore, we assume the initial models are sufficiently close to the optimal ones and that the underlying clusters are well-separated by a margin $\Delta$ (Assumption 6 in the appendix).

**Lemma 1.** *Let client $i$ belong to $S_j^t$ by ground-truth at the $t$-th communication round ($t \in [T]$). Assume that we obtain the cluster model $\theta_j^t$ such that there exist $0 < \beta < \frac{1}{2}$ and $\Delta := \min_{j \neq j'} \|\theta_j^* - \theta_{j'}^*\|$ that satisfy $\|\theta_j^t - \theta_j^*\| < (\frac{1}{2} - \beta)\sqrt{\frac{\lambda}{L}}\Delta$. Suppose we add zero-mean Gaussian noise with variance $\sigma_s^2$ to cluster identifiers with clipping bound 1. Then at the $t$-th round, the probability of misclassifying client $i$'s update into any other cluster $j' \neq j$ is upper bounded by*

$$\tau = \frac{8\eta^2 k}{\beta^2\lambda^2\Delta^4} + \frac{\sigma_s k}{\sqrt{\pi}}\exp(-1/4\sigma_s^2) + \frac{k^2\mu^2}{(M/k-B)^2}. \tag{1}$$

We note that in practice, the total number of clients is much larger than the number of clusters, i.e., $M/k \gg 1$. Hence $\tau$ can be much smaller than 1. In addition, $\tau$ decreases with the increase of the model separation parameter $\Delta$, which is expected. We prove Lemma 1 in Appendix C.2.

**Theorem 2.** *Assume in a certain iteration $t$ of Algorithm 2, we obtain $\theta_j$ such that $\|\theta_j - \theta_j^*\| < (\frac{1}{2} - \beta)\sqrt{\frac{\lambda}{L}}\Delta$ for some $\beta \in (0, \frac{1}{2})$. Denote $\theta_j^+$ to be the parameter in next iteration. We assume $B > 0$ so that each cluster receives a non-trivial number of (correctly assigned) clients after rebalancing. Then, for any $j \in [k]$, with probability at least $1 - \delta_c$, we have*

$$\|\theta_j^+ - \theta_j^*\| \leq \left(1 - \frac{\gamma L(B - 2\tau M/\delta_c)}{2\rho}\right)\|\theta_j - \theta_j^*\| + \epsilon,$$

$$\epsilon = \frac{4v}{\sqrt{\delta_c(B\delta_c - 4\tau M)}} + \frac{6\tau\gamma L\Delta M}{\delta_c B} + \frac{8\gamma vk\sqrt{\tau kM}}{B\delta_c\sqrt{\delta_c}} + \frac{2\gamma\sigma_\theta C_\theta}{B}\sqrt{d + 2\sqrt{d\ln(\frac{4}{\delta_c})} + 2\ln(\frac{4}{\delta_c})}. \tag{2}$$

We prove Theorem 2 in Appendix C.3. We note that this bound gets worse as the privacy noise $\sigma_\theta$ and $\sigma_s$ (embedded in $\tau$) increase. When $B = 0$, the rebalancing step is disabled and the algorithm reduces to the base (DP-)IFCA method, whose convergence can be directly analyzed in prior work; Theorem 2 specifically characterizes the rebalanced regime $B > 0$.

We note that this bound gets worse as the privacy noise $\sigma_\theta$ and $\sigma_s$ (embedded in $\tau$) increase. Additionally, note $\tau \propto \mathcal{O}(\Delta^{-4}M^{-2})$, then the error term $\epsilon$ with $\tau$ can be small if we have enough clients (large $M$) and a good cluster separation initialization (large $\Delta$). Further, we provide the final convergence results in Corollary 1 in the appendix. We discuss the effects of $B$ in the next paragraph.

**Discussions on the Bias/Variance Tradeoff.** Our method aims to achieve better tradeoffs between clustering bias and privacy noise variance by random rebalancing parameterized by $B$. The error term $\epsilon$ in Theorem 2 clearly exposes this relationship. Specifically, we can decompose $\epsilon$ into two parts: a bias term due to incorrect clustering and a variance term due to DP noise.

- **Bias term due to incorrect clustering:** The first three terms of $\epsilon$ depend on the incorrect clustering probability $\tau$, which increases as the clustering error $\tau$ grows. Since Lemma 1 shows $\tau$ is proportional to $\frac{1}{(M/k-B)^2}$, a larger $B$ could increases this bias. This aligns with the intuition that a larger $B$ indicates sampling more client updates into potentially wrong clusters.
- **Variance due to privacy noise:** The last term in $\epsilon$, $\mathcal{O}(\sigma_\theta/B)$, is directly proportional to the DP noise scale $\sigma_\theta$ and inversely proportional to $B$. This term decreases as $B$ increases.

As a result, the overall error $\epsilon$ shows how the clustering error and privacy noise affect model convergence. We observe that as $B$ increases, we accept potentially more biased cluster assignments in exchange for a significantly smaller privacy noise perturbation. This allows us to strike a better tradeoff between bias and variance, which we empirically demonstrate in Section 5.

## 5 EXPERIMENTS

In this section, we evaluate `RR-Cluster` across diverse datasets and scenarios to demonstrate its superior performance. We present main results in Section 5.2 on both image and text tasks, showing that our approach outperforms state-of-the-art DP clustering baselines in terms of privacy/utility tradeoffs and clustering quality. We also demonstrate the compatibility of `RR-Cluster` as a general plug-in to various federated clustering algorithms. We provide diagnostic analysis in Section 5.3.

### 5.1 EXPERIMENTAL SETTINGS

**Datasets.** We evaluate our method on four image datasets, one text dataset, and one synthetic dataset as follows. We use FashionMNIST (Xiao et al., 2017) that consists of 10-class 70,000 grayscale images of clothes, and EMNIST (Cohen et al., 2017), an extended dataset of the original MNIST, containing 814,255 images of handwritten characters with 47 categories covering both digits and letters. For the text data, we use the Shakespeare dataset (McMahan et al., 2017b; Shakespeare, 2014) on a next-character prediction task over the texts of Shakespeare's plays and writings. To control data separability and illustrate `RR-Cluster`'s behavior in non-private settings in more detail, we generate synthetic datasets for a linear regression task. Further, we also conduct experiments on both CIFAR10 and CIFAR100 (Krizhevsky et al., 2009) datasets; see Appendix E.1 for details.

To simulate different data distributions in real-world scenarios, we explore both manual partitions with a clear clustering structure as in previous works and partitions without clear cluster priors from previous FL benchmarks. For FashionMNIST, we followed the setting in IFCA (Ghosh et al., 2020), using image rotations to simulate the underlying ground truth clusters. We partition EMNIST based on a Dirichlet distribution over the labels (TFF), and partition the Shakespeare dataset based on speaking characters in the play (Caldas et al., 2018). In both image tasks, we have 1000 clients in total; for the Shakespeare dataset, we use 300 clients due to limitations of speaking roles. For all experiments, we use client sample rate $q = 0.1$, and tune $k \in \{2, 4\}$ based on validation data.

**Baselines.** We compare our methods with several competitive FL and federated clustering methods under DP: FedAvg (McMahan et al., 2017a), FedPer (Arivazhagan et al., 2019), IFCA (Ghosh et al., 2020), FeSEM (Long et al., 2023) and FedCAM (Ma et al., 2023). FedAvg is a classical FL baseline learning a global Model. IFCA, FeSEM and FedCAM are strong federated clustering approaches. While we focus on clustering (outputting $k$ models for $M$ clients where $k \ll M$) as opposed to broad federated personalization, we still compare with FedPer, a personalization method as a reference point.

We directly privatize these methods using standard subsampled Gaussian mechanisms guarantee the same client-level global DP across all approaches. For clustering methods, we privatize cluster identifiers similarly as in `RR-Cluster`. We do not compare with other personalization works that output $M$ models for $M$ clients.

## 5.2 COMPARISON WITH STRONG BASELINES

We present experiments comparing our method with other state-of-the-art methods on FashionMNIST, EMNIST, and Shakespeare datasets. For a fair comparison, we set the same hyperparameter (e.g., number of clients, local learning rates, local iterations, and communication rounds) for all methods. For DP settings, we tune the clipping bound to get optimal value for each task and each method separately. See Appendix D for additional details on experimental setup.

**Plugging `RR-Cluster` to Base Clustering Methods.** In Table 1, we use FashionMNIST rotation to create underlying clusters of the clients, following the setup in previous works (Ghosh et al., 2020; Kim et al., 2024) . For "Balanced Clusters", we first randomly divide the whole dataset into 1000 clients, and then group the clients evenly into four clusters as the ground truth. We rotate images in each cluster by 0, 90, 180, 270 degrees, respectively. For "Imbalanced Clusters", similarly, we rotate images (partitioned into three clusters at a ratio of 2:1:1) by 0, 90, 180 degrees. We plug `RR-Cluster` into existing federated clustering methods (IFCA (Ghosh et al., 2020), FeSEM (Long et al., 2023) and FedCAM) to demonstrate its compatibility. In both settings with balanced or imbalanced clusters, we see that `RR-Cluster` results in more significantly accurate final models compared with directly privatizing the base clustering algorithms without random rebalancing. We highlight improvement numbers for each setting in green; see convergence curves in Figure 2.

| Methods | Balanced Clusters | | | Imbalanced Clusters | | |
|---|---|---|---|---|---|---|
| | $\varepsilon = 2$ | $\varepsilon = 4$ | $\varepsilon = 8$ | $\varepsilon = 2$ | $\varepsilon = 4$ | $\varepsilon = 8$ |
| DP-FedAvg | 60.88 | 61.50 | 62.72 | 60.80 | 61.53 | 61.59 |
| DP-IFCA | 62.68 | 64.46 | 65.35 | 56.12 | 58.05 | 59.63 |
| DP-FeSEM | 63.16 | 64.68 | 64.76 | 58.55 | 59.82 | 60.10 |
| DP-FedCAM | 49.70 | 56.86 | 61.54 | 43.98 | 47.51 | 50.35 |
| RR-Cluster (IFCA) | 65.69 ↑3.01 | 66.63 ↑2.17 | 67.51 ↑2.16 | 61.31 ↑5.19 | 61.99 ↑3.94 | 63.77 ↑4.14 |
| RR-Cluster (FeSEM) | 63.89 ↑0.73 | 64.80 ↑0.12 | 64.86 ↑0.10 | 58.70 ↑0.15 | 60.34 ↑0.52 | 61.16 ↑1.06 |
| RR-Cluster (FedCAM) | 53.91 ↑4.21 | 58.65 ↑1.79 | 64.78 ↑3.24 | 46.12 ↑2.14 | 47.95 ↑0.44 | 50.78 ↑0.43 |

Table 1: Comparison with baselines on FashionMNIST. We validate the effectiveness of our method under various DP budget $\varepsilon$'s. `RR-Cluster` plugged into different clustering methods (the last panel) outperforms the corresponding baselines that directly privatize the base clustering algorithms without considering uncontrolled cluster sizes. Full results including comparisons with DP-FedPer are reported in Appendix E.3.

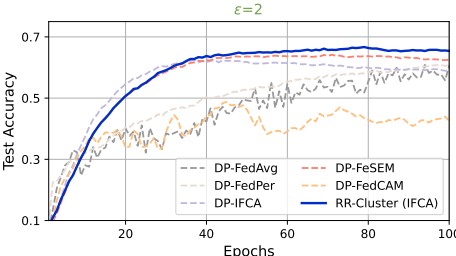 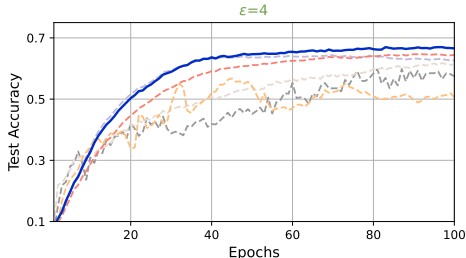

Figure 2: Convergence curves compared with strong baselines on FashionMNIST. We see that `RR-Cluster` achieves the better accuracies and faster convergence under both $\varepsilon$ values.

**Text Data with Ambiguous Cluster Patterns.** In Table 2, we study the performance on the Shakespeare dataset (Caldas et al., 2018), where each speaking role is a client. We do not control or manipulate the underlying clustering patterns a priori.

Our method outperforms DP-FedAvg, DP-IFCA, and other clustering baselines that directly integrate DP on this task as well. Note that '−' indicates DP-FedCAM method cannot converge in our experiments. This is due to that it requires clients to send both global model updates and cluster model updates, thus resulting in larger privacy noise. Meanwhile, the DP

| Methods | $\varepsilon = 4$ | $\varepsilon = 8$ | $\varepsilon = 16$ |
|---|---|---|---|
| DP-FedAvg | 04.47 | 13.43 | 17.53 |
| DP-IFCA | 12.62 | 12.64 | 13.20 |
| DP-FeSEM | 10.97 | 12.99 | 15.89 |
| DP-FedCAM | - | - | 04.47 |
| RR-Cluster (IFCA) | 13.42 | 13.83 | 16.25 |

Table 2: Comparison with baselines on Shakespeare. Our proposed approach outperforms the baselines in terms of test accuracies.

noise can be significantly reduced by incorporating
our `RR-Cluster` into FedCAM (Table 1).

**Varying Heterogeneity.** Here, we create datasets
with different degrees of heterogeneity. We partition the EMNIST data following a Dirichlet distribution with parameters $\alpha = \{0.5, 0.1\}$ for "Mild / High Heterogeneity" scenarios respectively. Results are reported in Table 3. We see that (1) most of the clustering methods outperform methods of training a global model in more heterogeneous setting, and (2) under different privacy $\varepsilon$ values, our method achieves highest accuracies in both heterogeneity settings.

| Methods | Mild Client Heterogeneity | | | High Client Heterogeneity | | |
|---|---|---|---|---|---|---|
| | $\varepsilon = 2$ | $\varepsilon = 4$ | $\varepsilon = 8$ | $\varepsilon = 2$ | $\varepsilon = 4$ | $\varepsilon = 8$ |
| DP-FedAvg | 26.47 | 30.95 | 32.61 | 23.82 | 24.46 | 25.54 |
| DP-IFCA | 62.23 | 65.39 | 65.93 | 32.53 | 35.60 | 36.19 |
| DP-FeSEM | 60.43 | 65.72 | 66.24 | 31.12 | 37.52 | 38.74 |
| DP-FedCAM | 57.35 | 57.22 | 61.26 | 23.66 | 24.55 | 29.70 |
| RR-Cluster (IFCA) | 66.38 ↑4.15 | 66.75 ↑1.03 | 67.04 ↑0.80 | 37.13 ↑4.60 | 38.92 ↑1.40 | 39.05 ↑0.31 |

Table 3: Comparison with baselines on the EMNIST dataset. We see that `RR-Cluster` outperforms other approaches especially when the privacy parameter is small (i.e., stronger privacy guarantees).

## 5.3 DIAGNOSTIC ANALYSIS

**Hyperparameter Analysis.** As discussed before, the rebalancing hyperparameter $B$ in our method introduces a tradeoff between privacy noise variance and clustering bias. When $B$ gets larger, more model updates can be sampled from large groups into small ones, leading to potentially larger clustering bias. On the other hand, when $B$ gets smaller, the privacy noise for small clusters may be large as the contributions of a single client in those groups are more difficult to hide. Choosing a proper value of $B$ ($0 < B < qM/k$) is critical for our algorithm. We run experiments on the FashionMNIST dataset, with the same image rotation settings in Section 5.2. We demonstrate the effects of $B$ in Figure 3. By changing the value of $B$, we can find an optimal value that leads to the best tradeoff. We find our method outperforms other private federated clustering baselines for a wide range of $B$'s.

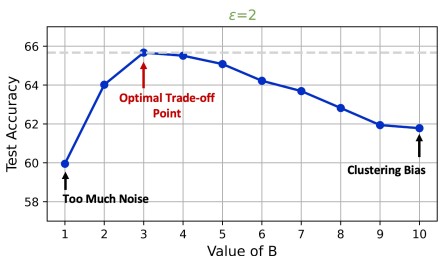

Figure 3: Ablation studies on the hyperparameter $B$ on the FashionMNIST dataset. There are a range of $B$'s that lead to improved performance. (Note that the baseline algorithm DP-IFCA in this setting achieves 62.68% accuracy.)

**Clustering Accuracy.** In addition to the performance of downstream tasks (i.e, learnt model accuracies) reported in previous sections, we also compare different methods in terms of clustering accuracies—the portion of clients that are correctly clustered. We experiment on FashionMNIST with balanced rotation as we have access to the underlying clustering structures. We present the success rate in Table 4. Our method achieves the highest clustering accuracy with the increase of $B$. Note that our method can exhibit higher clustering accuracies than the baselines even in non-private cases. This implies the seemingly incorrect assignment caused by expanding small clusters may actually help to stabilize clustering, mitigating model collapse, which often occurs in clustering (Ma et al., 2023) (also discussed Section 3.4). In the next paragraph, we illustrate this side effect on carefully-generated synthetic data in non-private cases.

| Methods | $\varepsilon = 0.5$ | $\varepsilon = 2$ | $\varepsilon = 4$ | $\varepsilon = 8$ | $\varepsilon = 16$ | $\varepsilon \to \infty$ |
|---|---|---|---|---|---|---|
| DP-IFCA | 34.37 | 42.18 | 39.06 | 40.62 | 42.18 | 75.00 |
| DP-FeSEM | - | 42.18 | 39.06 | 32.81 | 50.00 | 46.87 |
| RR-Cluster(IFCA) B=4 | 40.62 | 45.31 | 43.75 | 59.37 | 87.50 | 98.44 |
| RR-Cluster(IFCA) B=6 | 40.62 | 53.12 | 62.50 | 87.50 | 100.00 | 84.37 |
| RR-Cluster(IFCA) B=8 | **40.62** | **59.37** | **87.50** | **98.44** | **100.00** | **100.00** |

Table 4: Accuracy of clustering itself (i.e., percentage of correctly clustered clients) on FashionMNIST.

**Case Study on Synthetic Data.** To demonstrate `RR-Cluster`'s side effect of potentially mitigating model collapse, we conduct experiments on a synthetic dataset. The intuition is that by constraining the number of model updates that are assigned to any cluster is greater than $B$, we can ensure that each cluster model is at least trained using some related data, as opposed to never getting updated. In Appendix E.5, we show that in scenarios where baselines suffer from model collapse, `RR-Cluster` can successfully learn all $k$ clusters.

## 6 CONCLUSION

In this paper, we introduce `RR-Cluster`, a simple add-on that achieves better privacy/utility tradeoffs for federated clustering via a random sampling mechanism that controls client contributions across clusters, thereby reducing the effective DP noise. We have demonstrated the effectiveness of `RR-Cluster` both theoretically and empirically across various tasks.

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

# A   LLM Usage Disclosure

We used large language models (LLMs) solely to assist in polishing the writing of this manuscript such as improving grammar and correcting typos.

# B   Privacy Analysis

## B.1   RDP to DP Conversion

We restate existing RDP-to-DP conversion theorem, as follows.

**Theorem 3.** *(From RDP to $(\epsilon, \delta)$-DP (Mironov, 2017)) If mechanism $\mathcal{M}$ satisfies $(\lambda, \epsilon)$-RDP, then it also satisfies $(\epsilon + \frac{\log(1/\delta)}{\lambda - 1}, \delta)$-DP for any $\delta \in (0, 1)$.*

## B.2   Sensitivity Analysis

Continuing from Section 4.1, we study the sensitivity of our clustering updates each iteration for client-level DP. In global round $t$, we select $qM$ clients to update $k$ cluster models. Denote $\Delta\theta_i^t$ as the model updates from the $i$-th client at round $t$. By clipping, the $L_2$-norm of each update $\Delta\theta_i^t$ are limited within $[0, C_\theta]$. According to Algorithm 1, we add noise to the sum of cluster updates within each cluster. In the following, we investigate the $L_2$ sensitivity of the sum of cluster updates for any cluster under the client-level DP setup.

*Proof.* Suppose that we have two adjacent client sets, $\Theta$ and $\Theta'$, where $\Theta'$ has one more/less client than $\Theta^t$. We denote the added/removed update as $\Delta\theta_A$. We note that adding/removing a client may or may not affect other groups of model updates based on our algorithm. Without loss of generality, we assume the added/removed client update belongs to cluster 1. We discuss two cases as follows.

**Case 1**: Adding or removing one client belonging to the groups assigned to cluster 1 does not trigger random sampling and rebalancing. For instance, cluster 1 after removing one client still satisfies that the cardinality is at least $B$. For the sensitivity of the function outputting $k$ cluster's model updates, we only need to consider cluster 1, as all other groups remain unchanged. Denote the set of model updates assigned to cluster 1 before removing $\Delta\theta_A$ as $\{\Delta\theta_{1,1}, \cdots, \Delta\theta_{1,n_1}, \Delta\theta_A\}$. Then the sensitivity of summing up model updates in cluster 1 can be upper bounded by

$$\max_{\Theta, \Theta'} \|(\Delta\theta_{1,1} + ... + \Delta\theta_{1,n_1} + \Delta\theta_A) - (\Delta\theta_{1,1} + ... + \Delta\theta_{1,n_1})\| = \|\Delta\theta_A\| \leq C_\theta. \tag{3}$$

If we add $\Delta\theta_A$, the same sensitivity holds.

**Case 2**: The added/removed client triggers random sampling from large clusters to expand small ones. For example, removing a client results in cluster 1 originally having $B$ updates to only have $B - 1$ updates, in which case we would need to sample one model update from another cluster (denoted as $\Delta\theta'$) and insert it into cluster 1. The sensitivity of summing up model updates in cluster 1 is

$$\max_{\Theta, \Theta'} \|(\Delta\theta_{1,1} + ... + \Delta\theta_{1,n_1} + \Delta\theta') - (\Delta\theta_{1,1} + ... + \Delta\theta_{1,n_1} + \Delta\theta_A)\|$$
$$\leq \|\Delta\theta_A\| + \|\Delta\theta'\| \leq 2C_\theta. \tag{4}$$

Similarly, we have that the sensitivity corresponding to the cluster that we sample $\Delta\theta'$ from is $C_\theta$.

Hence, we prove the overall sensitivity is at most $2C_\theta$. Combining with the fact that we privatize each cluster model separately and $k$ new cluster model are obtained based on disjoint sets of clients, we can obtain desired privacy bounds. $\square$

## B.3   Privacy Guarantees

Here we provide the proof of Theorem 1. We first introduce the definition of Rényi differential privacy and several theorems used in our privacy analysis.

**Definition 3** (Rényi differential privacy (Mironov, 2017))**.** *A randomized mechanism $f : \mathcal{D} \to \mathcal{R}$ is said to have $\varepsilon$-Rényi differential privacy of order $\alpha$, or$(\alpha, \varepsilon)$-RDP for short, if for any adjacent*

$D, D' \in \mathcal{D}$ it holds that

$$D_\alpha(f(D)\|f(D')) = \frac{1}{\alpha - 1} \log \mathbb{E}_{x \sim f(D')} \left( \frac{f(D)}{f(D')} \right)^\alpha \le \varepsilon. \tag{5}$$

**Theorem 4** (RDP adaptive composition (Mironov, 2017))**.** *Let $f : \mathcal{D} \to \mathcal{R}_1$ be $(\alpha, \varepsilon_1)$-RDP and $g : \mathcal{R}_1 \times \mathcal{D} \to \mathcal{R}_2$ be $(\alpha, \varepsilon_2)$-RDP, then the mechanism defined as $(X, Y)$, where $X \sim f(D)$ and $Y \sim g(X, D)$, satisfies $(\alpha, \varepsilon_1 + \varepsilon_2)$-RDP.*

**Theorem 5** (RDP with Gaussian mechanism (Mironov, 2017))**.** *If $f$ has sensitivity $C$, then the Gaussian mechanism $G_\sigma f(D) = f(D) + N(0, C^2\sigma^2)$ satisfies $\left(\alpha, \frac{\alpha}{2\sigma^2}\right)$-RDP.*

**Theorem 6** (RDP with subsampling (Wang et al., 2019))**.** *Given a randomized mechanism $\mathcal{M}$, and let the randomized algorithm $\mathcal{M} \circ subsample$ be defined as: (1) subsample: subsample with subsampling rate $q$; (2) apply $\mathcal{M}$: a randomized algorithm taking the subsampled dataset as the input. For all integers $\alpha \ge 2$, if $\mathcal{M}$ is $(\alpha, \varepsilon)$-RDP, then $\mathcal{M} \circ subsample$ is $(\alpha, \varepsilon_q)$-RDP where*

$$\begin{aligned}
\varepsilon_q \le & \frac{1}{\alpha - 1} \log \left( 1 + q^2 \binom{\alpha}{2} \min \left\{ 4(e^{\varepsilon(2)} - 1), e^{\varepsilon(2)} \min \left[ 2, (e^{\varepsilon(\infty)} - 1)^2 \right] \right\} \right) \\
& + \sum_{j=3}^\alpha q^j \binom{\alpha}{j} e^{(j-1)\varepsilon(j)} \min \left\{ 2, (e^{\varepsilon(\infty)} - 1)^j \right\}.
\end{aligned} \tag{6}$$

In Theorem 6, DP parameter $\varepsilon$ is denoted as $\varepsilon(\alpha)$, since $\varepsilon$ can be viewed as a function of RDP parameter $\alpha$ for $1 \le \alpha \le \infty$ (Wang et al., 2019).

**Theorem 7** (Converting RDP into $(\varepsilon, \delta)$-DP (Mironov, 2017))**.** *If $f$ is an $(\alpha, \varepsilon)$-RDP mechanism, it also satisfies $(\varepsilon + \frac{\log 1/\delta}{\alpha - 1}, \delta)$-differential privacy for any $0 < \delta < 1$.*

We now provide the overall privacy guarantee of `RR-Cluster`, stated in Theorem 1. We analyze the privacy budget by considering three steps: (1) adaptive composition of private identifier and private model parameters in a single round, (2) privacy amplification with subsampling ratio $q$, and (3) composition over $T$ rounds.

We first provide the privacy guarantee for a single round `RR-Cluster` without subsampling. We restate the Proposition 1 in the main text and provide the proof.

*Proof.* According to Theorem 5, if using noise $\sigma_\theta = \sqrt{\frac{\alpha}{2\varepsilon_2}}$, we can calculate the privacy budget $\varepsilon$ as $\frac{\alpha}{2\sigma_\theta^2} = \varepsilon_2$. Which means for model parameters, it gives $(\alpha, \varepsilon_2)$-RDP. We then use adaptive RDP composition using Theorem 4 to compose budget $\varepsilon_1$ for identifiers and $\varepsilon_2$ for parameters to get $(\alpha, \varepsilon_1 + \varepsilon_2)$-RDP in total. $\qquad \square$

We then amplify the $(\alpha, \varepsilon_1 + \varepsilon_2)$-RDP budget with client subsampling with ratio $q$ and then compose it over $T$ rounds to provide overall privacy guarantee as follows. We denote $\varepsilon$ as $\varepsilon(\alpha)$ as in Theorem 6.

**Theorem 1.** *Suppose a randomized mechanism $\mathcal{M}$ that inputs a subset of selected clients $M^t$ and outputs $k$ new clustered models satisfies $(\alpha, \varepsilon_1(\alpha) + \varepsilon_2(\alpha))$-RDP for a single communication round of `RR-Cluster`. Given a client sampling ratio $q$ and total $T$ communication rounds, `RR-Cluster` algorithm satisfies $(\alpha, \varepsilon(\alpha))$-RDP where*

$$\begin{aligned}
\varepsilon(\alpha) \le & \frac{T}{\alpha - 1} \cdot \log \left( 1 + q^2 \binom{\alpha}{2} \min \left\{ 4(e^{\varepsilon_1(2)+\varepsilon_2(2)} - 1), \quad e^{\varepsilon_1(2)+\varepsilon_2(2)} \min\{2, (e^{\varepsilon_1(\infty)+\varepsilon_2(\infty)} - 1)^2\} \right\} \right. \\
& \left. + \sum_{j=3}^\alpha q^j \binom{\alpha}{j} e^{(j-1)(\varepsilon_1(j)+\varepsilon_2(j))} \min\{2, (e^{\varepsilon_1(\infty)+\varepsilon_2(\infty)} - 1)^j\} \right).
\end{aligned}$$

*Proof.* Given $(\alpha, \varepsilon_1 + \varepsilon_2)$-RDP in one round with sampling rate $q$, we can use Theorem 6 as

$$\begin{aligned}
\varepsilon(\alpha) \le & \frac{1}{\alpha - 1} \cdot \log \left( 1 + q^2 \binom{\alpha}{2} \min \left\{ 4(e^{\varepsilon_1(2)+\varepsilon_2(2)} - 1), \quad e^{\varepsilon_1(2)+\varepsilon_2(2)} \min\{2, (e^{\varepsilon_1(\infty)+\varepsilon_2(\infty)} - 1)^2\} \right\} \right. \\
& \left. + \sum_{j=3}^\alpha q^j \binom{\alpha}{j} e^{(j-1)(\varepsilon_1(j)+\varepsilon_2(j))} \min\{2, (e^{\varepsilon_1(\infty)+\varepsilon_2(\infty)} - 1)^j\} \right).
\end{aligned} \tag{7}$$

Then, we compose $\varepsilon_q$ over $T$ rounds still using adaptive composition as in Theorem 4 as

$$\varepsilon(\alpha) \leq \frac{T}{\alpha - 1} \cdot \log\left(1 + q^2\binom{\alpha}{2}\min\left\{4(e^{\varepsilon_1(2)+\varepsilon_2(2)} - 1), \quad e^{\varepsilon_1(2)+\varepsilon_2(2)}\min\{2, (e^{\varepsilon_1(\infty)+\varepsilon_2(\infty)} - 1)^2\}\right\}\right.$$

$$\left. + \sum_{j=3}^{\alpha} q^j\binom{\alpha}{j}e^{(j-1)(\varepsilon_1(j)+\varepsilon_2(j))}\min\{2, (e^{\varepsilon_1(\infty)+\varepsilon_2(\infty)} - 1)^j\}\right). \tag{8}$$

$\square$

Finally, we can convert the $(\alpha, \varepsilon(\alpha))$-RDP bound DP using Theorem 7. We have that for any $\delta \in (0, 1)$ (we used $0.001 < 1/M$ in our experiemtns), our algorithm satisfies $(\varepsilon, \delta)$-DP where

$$\varepsilon = \varepsilon(\alpha) + \frac{\log 1/\delta}{\alpha - 1}$$

$$\leq \frac{T}{\alpha - 1} \cdot \log\left(1 + q^2\binom{\alpha}{2}\min\left\{4(e^{\varepsilon_1(2)+\varepsilon_2(2)} - 1), \quad e^{\varepsilon_1(2)+\varepsilon_2(2)}\min\{2, (e^{\varepsilon_1(\infty)+\varepsilon_2(\infty)} - 1)^2\}\right\}\right.$$

$$\left. + \sum_{j=3}^{\alpha} q^j\binom{\alpha}{j}e^{(j-1)(\varepsilon_1(j)+\varepsilon_2(j))}\min\{2, (e^{\varepsilon_1(\infty)+\varepsilon_2(\infty)} - 1)^j\}\right). \tag{9}$$

## C CONVERGENCE ANALYSIS DETAILS

### C.1 ASSUMPTIONS

Here we state the assumptions used in our convergence analysis.

**Assumption 1.** *For every $j \in [k]$, $F^j(\cdot)$ is $\lambda$-strongly convex and $L$-smooth.*

**Assumption 2.** *For every $\theta$ and every $j \in [k]$, the variance of $f(\theta; z)$ is upper bounded by $\eta^2$, when $z$ is sampled from $\mathcal{D}_j$, i.e., $\mathbb{E}_{z \sim \mathcal{D}_j}\left[\left(f(\theta, z) - F^j(\theta)\right)^2\right] \leq \eta^2$.*

**Assumption 3.** *For every $\theta$ and every $j \in [k]$, the variance of $\nabla f(\theta; z)$ is upper bounded by $v^2$, where $z$ is sampled from $\mathcal{D}_j$, i.e., $\mathbb{E}_{z \sim \mathcal{D}_j}\left[\left\|\nabla f(\theta, z) - \nabla F^j(\theta)\right\|^2\right] \leq v^2$.*

**Assumption 4.** *For every cluster $S_j$ $(j \in [k])$ in any round, the variance of cluster sizes $|S_j|$ is upper bounded by $\mu^2$, i.e., $\mathbb{E}[(|S_j| - \frac{1}{k}\sum_{j \in [k]}|S_j|)^2] \leq \mu^2$.*

**Assumption 5.** *For every $\theta$ and every $j \in [k]$, the norm of the stochastic gradient $\nabla f(\theta; z)$ is bounded, i.e., there exists a constant $G > 0$ such that $\|\nabla f(\theta; z)\| \leq G, \forall z \sim \mathcal{D}_j$.*

Here, we further assume that the adopted DP clipping threshold $C_\theta$ is larger than any stocastic gradient norm, i.e., $C_\theta > G$, the clipping does not affect model update in our analysis. This assumption is commonly used across various differential privacy analysis (Li et al., 2022; Shi et al., 2023; Li et al., 2020b).

**Assumption 6.** *We assume that there exists $\alpha_0 \in (0, \frac{1}{2})$ s.t. the initial parameters $\theta_j^0$ satisfy $\|\theta_j^0 - \theta_j^*\| \leq (\frac{1}{2} - \alpha_0)\sqrt{\frac{\lambda}{L}}\Delta$ for every $j \in [k]$, where $\Delta := \min_{j \neq j'}\|\theta_j^* - \theta_{j'}^*\| \geq \mathcal{O}\left(\alpha_0^{\frac{1}{2}}k^{\frac{3}{4}}M^{-1}B^{-\frac{1}{2}}\right)$, $M > \left(\frac{B\Delta}{2\tau} - \frac{\rho}{\tau\gamma L}\right)$, and $\rho := M - (k-1)B$.*

Assumption 6 requires that the initialization models are good in the sense that they are close to the optimal cluster models. The condition on $\Delta$ indicates well-separated underlying cluster models. We note that the assumption on $M$ can be satisfied when the client population is large.

### C.2 PROOF OF LEMMA 1

*Proof.* Suppose we have cluster models $\{\theta_j\}_{j \in [k]}$, and assume that after certain steps, we have $\|\theta_j - \theta_j^*\| \leq (\frac{1}{2} - \beta)\sqrt{\frac{\lambda}{L}}\Delta, j \in [k]$. We first analyze the probability of incorrectly clustering a client taking into consideration the following three factors.

- Factor 1. Clients selecting their clusters by calculating training losses can introduce some inherent clustering error. We denote the probability of such incorrect clustering as $\Pr(E)$.

- Factor 2. Clients privatizing cluster identifiers introduces additional error, where the probability is denoted as $\Pr(H)$.

- Factor 3. `RR-Cluster` resamples updates from large to small groups, which also increases clustering error, denoted as $\Pr(O)$.

**Factor 1**: We begin with the definition of $E_i^{j,j'}$, which denotes an event where in a certain global round, client $i$ who belongs to cluster $j$ by ground truth, however been changed its cluster assignment to cluster $j'(j' \neq j)$, due to the min-loss cluster selection strategy. We bound this probability of erroneous cluster selection from Factor 1 as follows. Without loss of generality, we consider $E_i^{1,j}$. We have

$$\Pr\left(E_i^{1,j}\right) \leq \Pr\left(F_i(\theta_1) \geq F_i(\theta_j)\right) \leq \Pr\left(F_i(\theta_1) > t\right) + \Pr\left(F_i(\theta_j) \leq t\right) \tag{10}$$

for all $t \geq 0$. Choosing $t = \frac{F^1(\theta_1) + F^1(\theta_j)}{2}$, we have

$$\begin{aligned} \Pr(F_i(\theta_1) > t) &= \Pr\left(F_i(\theta_1) > \frac{F^1(\theta_1) + F^1(\theta_j)}{2}\right) \\ &= \Pr\left(F_i(\theta_1) - F^1(\theta_1) > \frac{F^1(\theta_j) - F^1(\theta_1)}{2}\right) \end{aligned} \tag{11}$$

for the first term. Considering that $F^1$ is $\lambda$-strongly convex and $L$-smooth and the assumption that $\|\theta_j^t - \theta_j^*\| \leq (\frac{1}{2} - \beta)\sqrt{\frac{\lambda}{L}}\Delta$ where $\Delta := \min_{j \neq j'} \|\theta_j^* - \theta_{j'}^*\|$, we have

$$F^1(\theta_j) \geq F^1(\theta_1^*) + \frac{\lambda}{2}\|\theta_j - \theta_1^*\|^2 \geq F^1(\theta_1^*) + \frac{\lambda\Delta^2}{2}(\frac{1}{2} + \beta)^2 \tag{12}$$

and

$$F^1(\theta_1) \leq F^1(\theta_1^*) + \frac{L}{2}\|\theta_1 - \theta_1^*\|^2 \leq F^1(\theta_1^*) + \frac{\lambda\Delta^2}{2}(\frac{1}{2} - \beta)^2. \tag{13}$$

Thus we have that

$$\frac{F^1(\theta_j) - F^1(\theta_1)}{2} \geq \frac{\lambda\Delta^2}{2}(\frac{1}{2} + \beta)^2 - \frac{\lambda\Delta^2}{2}(\frac{1}{2} - \beta)^2 = \frac{\beta\lambda\Delta^2}{2}. \tag{14}$$

According to the Chebyshev's inequality and the bounded variance assumption, we obtain $\Pr(F_i(\theta_1) > t) \leq \frac{4\eta^2}{\beta^2\lambda^2\Delta^4}$ and similarly, $\Pr(F_i(\theta_j) \leq t) \leq \frac{4\eta^2}{\beta^2\lambda^2\Delta^4}$, and thus $\Pr(E_i^{j,j'}) \leq \frac{8\eta^2}{\beta^2\lambda^2\Delta^4}$.

**Factor 2**: We analyze the probability that adding random Gaussian noise to the one-hot cluster identifiers changes the index with the maximum value. Let $\sigma_s^2$ denote the variance of the Gaussian noise. We consider the event $H_i^{j,j'}$, which means client $i$ is assigned to some cluster $j$ $(j \in [k])$ by the min-loss cluster selection strategy, but is changed into cluster $j'$ due to the added noise to guarantee differential privacy. In this step, we add Gaussian noise with variance $\sigma_s^2$ to each coordinate of the one-hot $s_i$, resulting in $\widetilde{s}_i$. Therefore,

$$\Pr(H_i^{j,j'}) \leq \Pr\left(1 + \mathcal{N}(0, \sigma_s^2) \leq 0 + \mathcal{N}(0, \sigma_s^2)\right) = \Pr(\mathcal{N}(0, 2\sigma_s^2) \leq -1). \tag{15}$$

Using the tail bound of Gaussian distributions $\Pr(x \geq t) \leq \frac{\sigma_s}{t\sqrt{2\pi}}\exp(-t^2/2\sigma_s^2)$, we have

$$\Pr(H_i^{j,j'}) \leq \frac{\sigma_s}{\sqrt{\pi}}\exp(-1/4\sigma_s^2). \tag{16}$$

**Factor 3**: We finally analyze the probability of event $O_i^{j,j'}$, where client $i$ is assigned to cluster $j$ after the previous two steps, but the assignment gets altered due to random rebalancing. Generally, if the mapping between a model update and a cluster has changed, then the model update should be in a large cluster $S_j, j \in [j_l]$. Assume that there are $h$ updates in a large cluster that are needed to merge into small clusters, we have

$$\Pr(O_i^{j,j'}) \leq \Pr(|S_j| > B) \cdot \frac{h}{\sum_{j \in j_l} |S_j|}. \tag{17}$$

For large clusters, considering the average cluster scale is $M/k$ by definition, and the variance of cluster scale is bounded by $\mu^2$. With one-sided Chebyshev's inequality, we can have

$$\Pr(|S_j| > B) \le \left(\frac{\mu}{M/k - B}\right)^2. \tag{18}$$

Combined with the facts that $h < M$ and $\sum_{j \in j_l} |S_j| > M/k$, it holds that

$$\Pr(O_i^{j,j'}) \le \left(\frac{\mu}{M/k - B}\right)^2 \cdot \frac{M}{M/k} = \frac{k\mu^2}{(M/k - B)^2}. \tag{19}$$

If the final cluster assignment is incorrect, then the error must have occurred in at least one of the three steps. Hence, we can bound the error clustering rate $\tau$ as

$$\tau_i^{j,j'} \le \Pr(E_i^{j,j'}) + \Pr(H_i^{j,j'}) + \Pr(O_i^{j,j'}) \le \frac{8\eta^2}{\beta^2 \lambda^2 \Delta^4} + \frac{\sigma_s}{\sqrt{\pi}} \exp(-1/4\sigma_s^2) + \frac{k\mu^2}{(M/k - B)^2}. \tag{20}$$

Note that $\tau_i^{j,j'}$ denotes the probability of client update $i$ being clustered into any single cluster $j'$. We can get the probability of any client update $i$ been wrongly clustered into any other cluster as

$$\tau = \bigcup_{j' \in [k]} \{\tau_i^{j,j'}\} \le k \cdot \tau_i^{j,j'} \le \frac{8\eta^2 k}{\beta^2 \lambda^2 \Delta^4} + \frac{\sigma_s k}{\sqrt{\pi}} \exp(-1/4\sigma_s^2) + \frac{k^2\mu^2}{(M/k - B)^2}. \tag{21}$$

$\square$

## C.3 PROOF OF THEOREM 2

*Proof.* Suppose that at a certain communication round, we have that $\|\theta_j - \theta_j^*\| \le (\frac{1}{2} - \beta)\sqrt{\frac{\lambda}{L}}\Delta$, for all $j \in [k]$. With out loss of generality, we focus on the update of cluster 1. Based on the updating rule, we have

$$\|\theta_1^+ - \theta_1^*\| = \left\|\theta_1 - \theta_1^* - \frac{\gamma}{|S_1|}\left(\sum_{i \in S_1} \nabla F_i(\theta_1) + \mathcal{N}(0, (2C_\theta)^2 \sigma_\theta^2 I_d)\right)\right\| \tag{22}$$

where $F_i(\theta) = F(\theta, D_i)$ with $D_i$ being the set of data from the $i$-th client used to compute the gradient. For cluster $j$, we denote its ground-truth client set as $S_j^*$, and the complement set as $\overline{S_j^*}$. Since $S_1 = (S_1 \cap S_1^*) \cup (S_1 \cap \overline{S_1^*})$, we have

$$\|\theta_1^+ - \theta_1^*\| = \|\theta_1 - \theta_1^* - \underbrace{\frac{\gamma}{|S_1|}\sum_{i \in S_1 \cap S^*} \nabla F_i(\theta_1)}_{T_1} - \underbrace{\frac{\gamma}{|S_1|}\sum_{i \in S_1 \cap \overline{S_i^*}} \nabla F_i(\theta_1)}_{T_2} - \underbrace{\frac{\gamma}{|S_1|}\mathcal{N}(0, (2\sigma_\theta C_\theta)^2 I_d)}_{T3}\|. \tag{23}$$

Using triangle inequality, we obtain $\|\theta_1^+ - \theta_1^*\| \le \|T_1\| + \|T_2\| + \|T_3\|$.

**Bound $\|T_1\|$:** We split $T_1$ as follows:

$$T_1 = \underbrace{\theta_1 - \theta_1^* - \widehat{\gamma}\nabla F^1(\theta_1)}_{T_{11}} + \underbrace{\widehat{\gamma}(\nabla F^1(\theta_1) - \frac{1}{|S_1 \cap S_1^*|}\sum_{i \in S_1 \cap S_1^*} \nabla F_i(\theta_1))}_{T_{12}}, \tag{24}$$

where $\widehat{\gamma} := \frac{\gamma |S_1 \cap S_1^*|}{|S_1|}$. Based on $\lambda$-strongly convexity and $L$-smoothness of $F^1$, we can bound $\|T_{11}\|$ as

$$\|T_{11}\| = \|\theta_1 - \theta_1^* - \widehat{\gamma}\nabla F^1(\theta_1)\| \le \left(1 - \frac{\widehat{\gamma}\lambda L}{\lambda + L}\right)\|\theta_1 - \theta_1^*\| \tag{25}$$

when $\widehat{\gamma} < \frac{1}{L}$. For $T_{12}$, we have $\mathbb{E}[\|T_{12}\|^2] = \frac{v^2}{|S_1 \cap S_1^*|}$, which means $E[\|T_{12}\|] \le \frac{v}{\sqrt{|S_1 \cap S_1^*|}}$. We can then bound the $\|T_{12}\|$ by Markov's inequality as: for any $\delta_1 \in (0, 1]$, we have with probability at least $1 - \delta_1$,

$$\|T_{12}\| \le \frac{v}{\delta_1\sqrt{|S_1 \cap S_1^*|}}. \tag{26}$$

Taking into accountant $T_{11}$, $T_{12}$ and fact that $\lambda < L$, we have with probability at least $1 - \delta_1$,

$$\|T_1\| \leq \left(1 - \frac{\gamma L |S_1 \cap S_1^*|}{2|S_1|}\right) \|\theta_1 - \theta_1^*\| + \frac{v}{\delta_1 \sqrt{|S_1 \cap S_1^*|}}. \tag{27}$$

**Bound $\|T_2\|$:** We propose to split the $(S_1 \cap \overline{S_1^*})$ into $\bigcup_{j \neq 1, j \in [k]} (S_1 \cap S_j^*)$. Without loss of generality, we first analyze the $(S_1 \cap S_j^*)$ as

$$T_{2j} = |S_1 \cap S_j^*| \underbrace{\nabla F^j(\theta_1)}_{T_{21j}} + \underbrace{\sum_{i \in S_i \cap S_j^*} (\nabla F_i(\theta_1) - \nabla F^j(\theta_1))}_{T_{22j}}, \tag{28}$$

where $T_2 = \frac{\gamma}{|S_1|} \sum_{j \in [k]} T_{2j}$ About $T_{21j}$, we have

$$\|T_{21j}\| = \|\nabla F^j(\theta_1) - \nabla F^j(\theta_j^*)\| \leq L\|\theta_1 - \theta_j^*\| \tag{29}$$
$$\leq L\|\theta_1 - \theta_1^*\| + L\|\theta_1^* - \theta_j^*\| \leq \frac{3}{2} L\Delta.$$

Further, we have $\mathbb{E}\left[\|T_{22j}\|^2\right] = |S_1 \cap S_j^*| v^2$, which implies $\mathbb{E}\left[\|T_{22j}\|\right] \leq \sqrt{|S_1 \cap S_j^*|} v$. And according to Markov's inequality, for any $\delta_2 \in [0, 1]$, with probability at least $1 - \delta_2$, we have

$$\|T_{22j}\| = \left\|\sum_{i \in S_1 \cap S_j^*} \nabla F_i(\theta_1) - \nabla F^j(\theta_1)\right\| \leq \frac{\sqrt{|S_1 \cap S_j^*|} v}{\delta_2}. \tag{30}$$

Considering there are $k$ clusters, we have with probability at least $(1 - k\delta_2)$,

$$\|T_2\| \leq \frac{3\gamma L\Delta}{2|S_1|} |S_1 \cap \overline{S_1^*}| + \frac{\gamma v \sqrt{k}}{\delta_2 |S_1|} \sqrt{|S_1 \cap \overline{S_1^*}|}. \tag{31}$$

**Bound $\|T_3\|$:** We give the norm bound of the gaussian noise from DP as follows. $\|T_3\| = \|\frac{\gamma}{|S_1|} \mathcal{N}(0, (2\sigma_\theta C_\theta)^2 I_d)\|$, which is a vector with dimension of model parameters $d$ and scale of DP noise multiplier. First, we have

$$\|T_3\| = \left\|\frac{\gamma}{|S_1|} \mathcal{N}(0, (2\sigma_\theta C_\theta)^2 I_d)\right\| \leq \frac{2\gamma \sigma_\theta C_\theta}{B} \|\mathcal{N}(0, I_d)\|. \tag{32}$$

Denote $X \sim \mathcal{N}(0, I_d)$. Then $\|X\|^2$ follows a $\chi^2$-distribution with $d$ degrees of freedom, i.e. $\|X\|^2 \sim \chi_d^2$. We then use the upper bound for a $\chi^2$-tail to bound $\|T_3\|$ as

$$P\left(\|X\|^2 \leq d + 2\sqrt{d \ln(\frac{1}{\delta_3})} + 2\ln(\frac{1}{\delta_3})\right) \geq 1 - \delta_3. \tag{33}$$

Multiplying both sides by $(\sigma_\theta / B)$ and taking the square root yields

$$P\left(\|T_3\| \leq \frac{2\gamma \sigma_\theta C_\theta}{B} \sqrt{d + 2\sqrt{d \ln(\frac{1}{\delta_3})} + 2\ln(\frac{1}{\delta_3})}\right) \geq 1 - \delta_3. \tag{34}$$

Which means that we have probability at least $1 - \delta_3$ that

$$\|T_3\| \leq \frac{2\gamma \sigma_\theta C_\theta}{B} \sqrt{d + 2\sqrt{d \ln(\frac{1}{\delta_3})} + 2\ln(\frac{1}{\delta_3})}. \tag{35}$$

With bounded $\|T_1\|$ and $\|T_2\|$ above, we have probability at least $1 - (\delta_1 + k\delta_2 + \delta_3)$ that

$$\|\theta_1^+ - \theta_1^*\| \leq \left(1 - \frac{\gamma L |S_1 \cap S_1^*|}{2|S_1|}\right) \|\theta_1 - \theta_1^*\|$$
$$+ \frac{v}{\delta_1 \sqrt{|S_1 \cap S_1^*|}} + \frac{3\gamma L\Delta}{2|S_1|} |S_1 \cap \overline{S_1^*}| \tag{36}$$
$$+ \frac{\gamma v \sqrt{k}}{\delta_2 |S_1|} \sqrt{|S_1 \cap \overline{S_1^*}|} + \frac{2\gamma \sigma_\theta C_\theta}{B} \sqrt{d + 2\sqrt{d \ln(\frac{1}{\delta_3})} + 2\ln(\frac{1}{\delta_3})}.$$

We bound $|S_1|$, $|S_1 \cap S_1^*|$ and $|S_1 \cap \overline{S_1^*}|$ next to complete the theorem. Based on the updating rules of our method, we note that the maximum cardinality of any $S_j$ is bounded by $\rho := M - (k-1)B$, thus we have $|S_1| \le \rho$. Also, after rebalancing, we have $|S_1| \ge B$. Given the error rate in clustering from Lemma 1, we have $\mathbb{E}[|S_1 \cap \overline{S_1^*}|] \le \tau M$, plus the Markov's inequality, we obtain probability at least $1 - \delta_4$ that $|S_1 \cap \overline{S_1^*}| \le \tau M/\delta_4$, meanwhile, we have $|S_1 \cap S_1^*| = |S_1| - (|S_1 \cap \overline{S_1^*}|) \ge (B - \frac{\tau M}{\delta_4})$. Let $\delta_c \ge \delta_1 + k\delta_2 + \delta_3 + \delta_4$ and choose $\delta_1 = \delta_c/4, \delta_2 = \delta_c/4k, \delta_3 = \delta_c/4, \delta_4 = \delta_c/4$, then the failure probability can be bounded by $\delta_c$. Finally, we have with probability at least $(1 - \delta_c)$, it holds that

$$\|\theta_1^+ - \theta_1^*\| \le \left(1 - \frac{\gamma L(B - \frac{2\tau M}{\delta_c})}{2\rho}\right) \|\theta_1 - \theta_1^*\|$$

$$+ \frac{4v}{\sqrt{\delta_c(B\delta_c - 4\tau M)}} + \frac{6\tau\gamma L\Delta M}{\delta_c B} + \frac{8\gamma vk\sqrt{\tau kM}}{B\delta_c\sqrt{\delta_c}} + \frac{2\gamma\sigma_\theta C_\theta}{B}\sqrt{d + 2\sqrt{d\ln(\frac{4}{\delta_c})} + 2\ln(\frac{4}{\delta_c})}, \tag{37}$$

where

$$\rho = (M - (k-1)B), \text{ and } \tau = \frac{8\eta^2 k}{\beta^2\lambda^2\Delta^4} + \frac{\sigma_s k}{\sqrt{\pi}}\exp(-1/4\sigma_s^2) + \frac{k^2\mu^2}{(M/k - B)^2}. \tag{38}$$

$\square$

## C.4 Overall Convergence of RR-Cluster over $T$ Rounds

**Corollary 1.** *Suppose Assumption 1-6 hold, after* $T = \frac{1}{K}\log\left(\frac{\Delta\sqrt{\lambda}}{2\epsilon_T\sqrt{L}}\right) + \frac{\log\left(\frac{\Delta\sqrt{\lambda}}{8(\frac{1}{2}-\alpha_0)\sqrt{L}}\right)}{\log(1-K)}$ *rounds, for all cluster* $j \in [k]$, *we have at least probability* $(1 - \delta_c')$, $\|\theta_j^T - \theta_j^*\| \le \epsilon_T$, *where*

$$\epsilon_T = \frac{2\epsilon}{K}, \ K = \frac{\gamma L(B - 2\tau M/\delta_c)}{2\rho}, \ \delta_c' = kT\delta_c.$$

*Proof.* Recall that the error floor of Theorem 2 $\epsilon$ is

$$\epsilon = \frac{4v}{\sqrt{\delta_c(B\delta_c - 4\tau M)}} + \frac{6\tau\gamma L\Delta M}{\delta_c B} + \frac{8\gamma vk\sqrt{\tau kM}}{B\delta_c\sqrt{\delta_c}} + \frac{2\gamma\sigma_\theta C_\theta}{B}\sqrt{d + 2\sqrt{d\ln(\frac{4}{\delta_c})} + 2\ln(\frac{4}{\delta_c})}. \tag{39}$$

Let $K$ be the subtraction factor $\frac{\gamma L(B - 2\tau M/\delta_c)}{2\rho}$ in Eq. equation 37. We rewrite Eq. equation 37 as

$$\|\theta_1^+ - \theta_1^*\| \le (1 - K)\|\theta_1 - \theta_1^*\| + \epsilon = \|\theta_1 - \theta_1^*\| + (\epsilon - K\|\theta_1 - \theta_1^*\|). \tag{40}$$

Given the bound of $\Delta$ in Assumption 6, we can ensure $\epsilon \le K(\frac{1}{2} - \alpha_0)\Delta$, thus we have that the proposed method is contractive, that is, $\|\theta_1^{t+1} - \theta_1^*\| \le \|\theta_1^t - \theta_1^*\|$.

In the $t$-th round, assume we have $\|\theta_1^t - \theta_1^*\| \le (\frac{1}{2} - \alpha_t)\sqrt{\frac{\lambda}{L}}\Delta$, where the sequence of $\alpha_t$ should be non-decreasing due to its contractive convergence.

Suppose after $T'$ rounds we have $\alpha_t \le \frac{1}{4}$, this can be achieved if both

$$(1 - K)^{T'}(\frac{1}{2} - \alpha_0)\Delta \le \frac{1}{8}\sqrt{\frac{\lambda}{L}}\Delta, \text{ and } \frac{1}{K}\epsilon \le \frac{1}{8}\sqrt{\frac{\lambda}{L}}\Delta \tag{41}$$

hold, where the latter equation holds given bounds on $\Delta$ in Assumption 6. Solving Eq. equation 41, we can get

$$T' \ge \frac{\log\left(\frac{\Delta\sqrt{\lambda}}{8(\frac{1}{2}-\alpha_0)\sqrt{L}}\right)}{\log(1 - K)}, \tag{42}$$

which means that after $T'$ rounds, $\|\theta_1^{T'} - \theta_1^*\| \le \frac{1}{4}\sqrt{\frac{\lambda}{L}}\Delta$.

Let $T = T' + T''$, after $T$ rounds in total (with another $T''$ rounds), we have probability at least $1 - kT\delta_c$, for all $j \in [k]$

$$\|\theta_1^T - \theta_1^*\| \le (1 - K)^{T''}|\theta_1^{T'} - \theta_1^*| + \frac{1}{K}\epsilon, \tag{43}$$

where the last term $\frac{\epsilon}{K}$ is calculated by the sum of series of $\epsilon + (1-K)\epsilon + (1-K)^2\epsilon + \cdots$.

Let $T'' \geq \frac{1}{K} \log \left( \frac{k\Delta\sqrt{\lambda}}{4\epsilon\sqrt{L}} \right)$, we can bound $(1-K)^{T''} \|\theta_1^{T'} - \theta_1^*\|$ as

$$(1-K)^{T''} \|\theta_1^{T'} - \theta_1^*\| \leq e^{-KT''} \cdot \frac{1}{4}\sqrt{\frac{\lambda}{L}}\Delta \leq \frac{1}{K}\epsilon, \tag{44}$$

which means $\|\theta_1^T - \theta_1^*\| \leq \frac{2\epsilon}{K} = \epsilon_T$ after $T = \frac{1}{K} \log \left( \frac{K\Delta\sqrt{\lambda}}{4\epsilon\sqrt{L}} \right) + \frac{\log\left( \frac{\Delta\sqrt{\lambda}}{8(\frac{1}{2}-\alpha_0)\sqrt{L}} \right)}{\log(1-K)} = \frac{1}{K} \log \left( \frac{\Delta\sqrt{\lambda}}{2\epsilon_T\sqrt{L}} \right) + \frac{\log\left( \frac{\Delta\sqrt{\lambda}}{8(\frac{1}{2}-\alpha_0)\sqrt{L}} \right)}{\log(1-K)}$ rounds, with probability at least $(1-\delta_c')$ where $\delta_c' = kT\delta_c$.

$\square$

# D    EXPERIMENTAL DETAILS

**Models.**    For both image classification tasks, we employed a CNN network, and for the Shakespeare classification task, we used a LSTM network for classification. For the synthetic data, we used a linear regression model with slope and interscet as trainable parameters.

**Hyperparameters.**    For the clipping bound of model updates $C_\theta$, we conduct grid search in `np.logspace(-1,-3,5)` and select the value based on model accuracy on validation set for all methods. For our hyperparameter $B$ (minimal cluster size), if not specified, we use grid search $B$ in $\{4, 8, 12\}$ by default in all experiments. For the clipping threshold of cluster identifiers $s$, considering that all $s$ are one-hot vectors, the clipping threshold cannot affect cluster assignment. Since $\widetilde{s} = s/\max(1, \frac{s}{C_s}) + \mathcal{N}(0, C_s^2) = C_s \cdot (s + \mathcal{N}(0,1))$ if we select $C_s \leq 1$, and that server assign clusters based on the position of $\max\{\widetilde{s}_j\}_{j\in k}$, so it's only the position of $\max\{\widetilde{s}_j\}_{j\in k}$ that matters, which is irrelevant to $C_s$. In experiments, we use $0.1$ as the $C_s$. We tune local client-side learning rate from {1e-6, 1e-5, 1e-4, 1e-3}, and the number of local epochs from {5, 10, 15, 20} on DP-FedAvg McMahan et al. (2018) and use the same values for all methods.

**Hardware.**    Most experiments are conducted using 2 NVIDIA L40 GPU on a AMD Thread-Ripper HEDT platform in a desktop.

# E    ADDITIONAL EXPERIMENTAL RESULTS

## E.1    EXPERIMENTS ON CIFAR10, CIFAR100 AND TINYIMAGENET

We further evaluate our method `RR-Cluster` on the CIFAR10 and CIFAR100 datasets (Krizhevsky et al., 2009). For both datasets, we partition the dataset in the same method as "Balanced Clusters" setup for FashionMNIST described in Section 5.2. We compare `RR-Cluster` with various DP federated learning and federated clustering baselines. The experiments are conducted with privacy budgets $\varepsilon = 4$ and $\varepsilon = 8$. We present classification accuracies in Table 6. As shown, RR-Cluster consistently outperforms the baseline methods on both CIFAR10 and CIFAR100 datasets.

## E.2    FINE-GRAINED ACCURACY METRICS WITH MAX/MIN ACCURACY

To provide a more comprehensive breakdown of model behavior under differential privacy, we present a unified table containing *average*, *maximum*, and *minimum* client accuracy in Table 5.

| Methods | $\varepsilon = 2$ | | | $\varepsilon = 4$ | | | $\varepsilon = 8$ | | |
|---|---|---|---|---|---|---|---|---|---|
| | Avg | Max | Min | Avg | Max | Min | Avg | Max | Min |
| DP-FedAvg | 60.88 | 69.20 | 41.94 | 61.50 | 69.38 | 45.71 | 62.72 | 71.14 | 48.65 |
| DP-IFCA | 62.68 | 75.85 | 08.82 | 64.46 | 76.73 | 21.00 | 65.35 | 79.55 | 22.12 |
| DP-FeSEM | 63.16 | 74.40 | 16.12 | 64.68 | 77.76 | 18.42 | 64.76 | 78.44 | 24.32 |
| DP-FedCAM | 49.70 | 57.74 | 18.33 | 56.86 | 66.48 | 21.43 | 61.54 | 72.32 | 29.44 |
| RR-Cluster (IFCA) | 65.69 | 82.86 | 28.42 | 66.63 | 81.12 | 37.83 | 67.51 | 84.38 | 43.24 |

Table 5: Fine-grained comparison of client-wise performance, including average, maximum and minimum client accuracy under different privacy budgets.

**Discussion.** By consolidating average, maximum, and minimum accuracy into a single table, we highlight how RR-Cluster improves client-wise performance across a wide spectrum of client conditions.

| Methods | CIFAR10 | | CIFAR100 | | TinyImagenet | |
|---|---|---|---|---|---|---|
| | $\varepsilon = 4$ | $\varepsilon = 8$ | $\varepsilon = 4$ | $\varepsilon = 8$ | $\varepsilon = 4$ | $\varepsilon = 8$ |
| DP-FedAvg | 52.89 | 54.48 | 14.05 | 14.49 | 13.67 | 14.06 |
| DP-FedProx | 52.04 | 54.61 | 14.48 | 14.53 | 14.01 | 14.12 |
| DP-IFCA | 53.91 | 56.77 | 16.29 | 17.47 | 15.79 | 16.76 |
| DP-FeSEM | 53.01 | 55.87 | 16.23 | 17.08 | 15.65 | 16.44 |
| DP-FedCAM | 52.50 | 54.73 | 16.36 | 17.04 | 15.71 | 16.38 |
| RR-Cluster (IFCA) | **57.46** | **58.41** | **17.83** | **18.00** | **17.14** | **17.90** |

Table 6: Comparison with baselines on CIFAR10, CIFAR100 and TinyImagenet datasets. RR-Cluster achieves the highest accuracies.

### E.3 FULL RESULTS WITH FEDERATED PERSONALIZATION METHODS

| Methods | Balanced Clusters | | | Imbalanced Clusters | | |
|---|---|---|---|---|---|---|
| | $\varepsilon = 2$ | $\varepsilon = 4$ | $\varepsilon = 8$ | $\varepsilon = 2$ | $\varepsilon = 4$ | $\varepsilon = 8$ |
| DP-FedAvg | 60.88 | 61.50 | 62.72 | 60.80 | 61.53 | 61.59 |
| DP-FedPer | 61.03 | 61.78 | 62.19 | 60.42 | 61.83 | 62.35 |
| DP-IFCA | 62.68 | 64.46 | 65.35 | 56.12 | 58.05 | 59.63 |
| DP-FeSEM | 63.16 | 64.68 | 64.76 | 58.55 | 59.82 | 60.10 |
| DP-FedCAM | 49.70 | 56.86 | 61.54 | 43.98 | 47.51 | 50.35 |
| RR-Cluster (IFCA) | 65.69 $_{\uparrow 3.01}$ | 66.63 $_{\uparrow 2.17}$ | 67.51 $_{\uparrow 2.16}$ | 61.31 $_{\uparrow 5.19}$ | 61.99 $_{\uparrow 3.94}$ | 63.77 $_{\uparrow 4.14}$ |
| RR-Cluster (FeSEM) | 63.89 $_{\uparrow 0.73}$ | 64.80 $_{\uparrow 0.12}$ | 64.86 $_{\uparrow 0.10}$ | 58.70 $_{\uparrow 0.15}$ | 60.34 $_{\uparrow 0.52}$ | 61.16 $_{\uparrow 1.06}$ |
| RR-Cluster (FedCAM) | 53.91 $_{\uparrow 4.21}$ | 58.65 $_{\uparrow 1.79}$ | 64.78 $_{\uparrow 3.24}$ | 46.12 $_{\uparrow 2.14}$ | 47.95 $_{\uparrow 0.44}$ | 50.78 $_{\uparrow 0.43}$ |

Table 7: Full version of Table 1. Comparison with baselines on FashionMNIST. See discussions in Section 5.2.

In this section, we present the full versions of the experimental results tables discussed in Section 5.2 of the main paper. For detailed analysis and discussion of these findings, please refer to Section 5.2.

### E.4 ACCURACY COMPARISON WITH OTHER BASELINES

The re-balancing operation mixes small cluster updates and other updates together, which makes the cluster model actually move closer to the conceptual 'global average model' (which actually doesn't exist in federated clustering). We compare our RR-Cluster with a baseline which directly adopts the Global Model For Small Clusters as GMFSC in Table 10. We conduct the experiments with the 'Balanced Clusters' setting on FashionMNIST dataset. Our hyperparameters remains the same as that

| Methods | Mild Client Heterogeneity | | | High Client Heterogeneity | | |
|---|---|---|---|---|---|---|
| | $\varepsilon = 2$ | $\varepsilon = 4$ | $\varepsilon = 8$ | $\varepsilon = 2$ | $\varepsilon = 4$ | $\varepsilon = 8$ |
| DP-FedAvg | 26.47 | 30.95 | 32.61 | 23.82 | 24.46 | 25.54 |
| DP-FedPer | 40.67 | 40.81 | 41.23 | 24.78 | 25.51 | 26.12 |
| DP-IFCA | 62.23 | 65.39 | 65.93 | 32.53 | 35.60 | 36.19 |
| DP-FeSEM | 60.43 | 65.72 | 66.24 | 31.12 | 37.52 | 38.74 |
| DP-FedCAM | 57.35 | 57.22 | 61.26 | 23.66 | 24.55 | 29.70 |
| RR-Cluster (IFCA) | 66.38 ↑4.15 | 66.75 ↑1.03 | 67.04 ↑0.80 | 37.13 ↑4.60 | 38.92 ↑1.40 | 39.05 ↑0.31 |

Table 8: Full version of Table 3. Comparison with baselines on EMNIST. See discussions in Section 5.2.

| Methods | $\varepsilon = 4$ | $\varepsilon = 8$ | $\varepsilon = 16$ |
|---|---|---|---|
| DP-FedAvg | 04.47 | 13.43 | 17.53 |
| DP-FedPer | 13.09 | 13.14 | 13.18 |
| DP-IFCA | 12.62 | 12.64 | 13.20 |
| DP-FeSEM | 10.97 | 12.99 | 15.89 |
| DP-FedCAM | - | - | 04.47 |
| RR-Cluster(IFCA) | 13.42 | 13.83 | 16.25 |

Table 9: Full version of Table 2. Comparison with baselines on Shakespeare. See discussions in Section 5.2.

in Section 5.2. It shows that the RR-Cluster ourperform the GMFSC baseline, due to aggregated updates with partial small cluster updates works suitable for small cluster itself, instead to general knowledge from the average model.

| Methods | Balanced Clusters | | |
|---|---|---|---|
| | $\varepsilon = 2$ | $\varepsilon = 4$ | $\varepsilon = 8$ |
| GMFSC | 64.62 | 66.45 | 67.23 |
| RR-Cluster (IFCA) | 65.69 | 66.63 | 67.51 |

Table 10: Comparing RR-Cluster with "global model for small cluster" baseline (GMFSC) See discussions in Appendix E.4.

## E.5 EXPERIMENTS ON SYNTHETIC DATA

To further demonstrate that RR-Cluster has the side effect of mitigating model collapse in non-private settings, we conduct experiments on a synthetic dataset. Each synthetic dataset pair $(x, \hat{y})$ with $x, \hat{y} \in \mathbb{R}$ is generated by $\hat{y} = kx + b + \epsilon$, where $\epsilon \sim \mathcal{N}(0, \sigma^2)$. The trainable parameters are $k, b \in \mathbb{R}$, and we use the MSE loss. We create four scenarios based on whether the clusters are balanced, and whether there is a clear separation between clusters.

When the truth number of clusters is large (first row), we see that IFCA suffers from model collapse even if we set $k = 4$. In contrast, our RR-Cluster efficiently trained all four cluster models. We also note that in the other scenario where the clusters are imbalanced (so that it is more likely to dynamically rebalance the size of clusters) (second row), our method can introduce some additional bias compared with IFCA. Further, we extend the original data distribution (the 'Easy to Cluster') to 'Hard to Cluster' settings by a increased Gaussian variance on client data generation. We show that our RR-Cluster works well with handling data that harder to cluster, and that our methods also demonstrate strong robustness against model collapse, since we have four cluster model fully aligned with the object data, while IFCA have one cluster model left at the bottom and remained untrained.

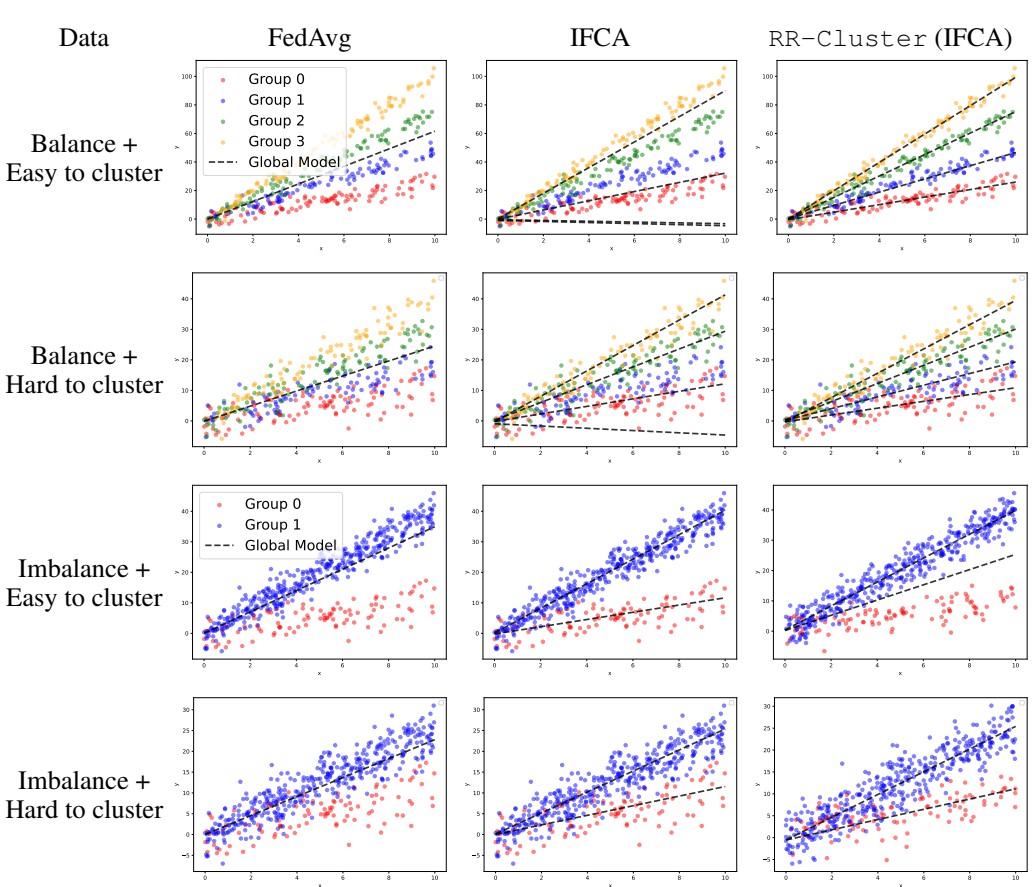

Table 11: To understand the effects of RR-Cluster in non-private settings, we simulate the balanced and imbalanced distribution of clients on the synthetic dataset and compare with other methods. The original datapoints (data generated from different noisy line are in different colors) is shown in each pictures. We plot the final global/cluster model as black dashlines in each picture.

