# OpenReview forum: "Differentially Private Federated Clustering with Random Rebalancing"
_ICLR.cc/2026/Conference — Submitted to ICLR 2026_

### Official Review · Reviewer_NBjo · 2025-10-30

**Soundness:** 2
**Presentation:** 2
**Contribution:** 2
**Rating:** 2
**Confidence:** 3

**Summary:**

This paper considers the problem of differential private federated clustering. The imbalanced cluster sizes leads to noise that may hurt utility of models. Then, this paper propose to use a general trick called random rebalancing cluster (RR Cluster) that downsamples large clusters to a fixed size to have more balanced clusters and thus better noise designs. Theoretical convergence rates are given in the paper. Empirical results suggest that RR-Cluster can significantly improve privacy / utility tradeoffs.

**Strengths:**

- Originality: The clustering algorithms and differential privacy mechanisms remain unchanged, the originality of RR-cluster that down sample large cluster is okish but not surprising.
- Significance: having a plug-and-play trick is good and could be practical but may not be significant enough.
- Clarity: The paper is mostly clear for readers who have some knowledge of the literature.
- Quality: The motivation looks reasonable.

**Weaknesses:**

- The overall convergence rate of Algorithm 2 (RR-Cluster-IFCA) is left in the appendix (Corollary 1) and the rates seem to be inconsistent with the baseline rates of IFCA (Checkout the Corollary 2 of the original IFCA paper) even when we ignore the  differential privacy part. I hope the authors can explain the difference.

- The Theorem 2 is not good enough.
    -  It is claimed in Line 53 that having $B=0$, the RR-Cluster type algorithm degenerate to base clustering algorithms. However, the upper bound of Theorem 2 goes to infinity as B shrinks to 0.

- Writings can be improved:
    - $F^j$ is used to denote both empirical and population loss between line 138 and 143.
    - $\hat{\theta}^*$ looks kind of strange.
    - When quantities, such as $q$, appear in the paper for the first time, are not explained (such as domain) which leads to some difficulty in reading.
    - Algorithm 1 is not presented in a clear way. For example, line 5 says "some information $s_i$", and line 6 says "privatizes $\{s_i\}$ into $\{\tilde{s}_i\}$". It is clearly better to articulate what are these information and perhaps reference to some equations that explain them.

**Questions:**

See the weakness.

---

> ### Author Response · Authors · 2025-11-21
>
> We thank the reviewer for the comments.
>
> **[W1: Convergence rate different from IFCA]** Our RR-Cluster(IFCA) bound is derived in a strictly harder setting than the original IFCA analysis: we add (1) user-level DP noise on both model updates and cluster indicators, and (2) a random rebalancing step controlled by B. These two mechanisms introduce additional error terms that do not exist in IFCA.
>
> A key consequence is that our assignment-error term $\tau$ in Lemma 1 cannot be bounded by constants with high probability, unlike IFCA Eq. (23–24). So the $\tau$ related terms and DP terms emerge inevitably in the final convergence rate. This is exactly why the contraction factor and error floor in Theorem 2 contain additional B components.
>
> **[W2: Concerns about Theorem 2]**
> We want to emphasize the broader intention behind our theoretical contribution in Theorem 2. Our analysis is not meant to mirror the original IFCA rate, but to depict the different dynamics that arise from user-level DP noise and the rebalancing. RR-Cluster introduces differences from the non-private baseline method in three essential ways (i.e., DP noise on cluster assignments, DP noise for parameter updates, and enforced minimum clusters), and these mechanisms necessarily lead to additional error terms in the contraction behavior.
>
> More importantly, in Theorem 2, we reveal the key tradeoff that motivates the method: increasing B reduces DP noise via averaging but increases potential clustering bias. Quantifying this bias/variance tradeoff is one of the main purposes of Theorem 2 and Corollary 1.
>
>
> **[W3: Writings]**
> Thanks for the suggestions. We have fixed the typos and clarified Algorithm 2 in the revision.  Specifically, 1) we fixed the use of $F^j$; 2) we removed the hat and turn to use $\theta_j$ to represent the cluster models; 3) we explained the meaning of q and other quantities before using them; 4) we added the explanation of $s_i$ and references in Algorithm 1.

---

### Official Review · Reviewer_ax89 · 2025-10-31

**Soundness:** 3
**Presentation:** 3
**Contribution:** 3
**Rating:** 6
**Confidence:** 3

**Summary:**

This work proposes RR-Cluster, a method designed to balance client contributions in differentially private federated clustering. It ensures that each cluster includes contributions from at least $B$ clients, thereby reducing the impact of differential privacy noise on small clusters compared with existing approaches. The authors provide both privacy and convergence analyses of RR-Cluster, and its effectiveness is further demonstrated through empirical experiments.

**Strengths:**

1. The paper is clearly written and easy to follow.

2. The proposed RR-Cluster method is promising, as increasing the number of clients contributing to small clusters effectively mitigates the noise intensity introduced by differential privacy.

3. The authors conduct both theoretical (privacy and convergence) and empirical evaluations of RR-Cluster. The derived theoretical bounds also capture the bias–variance trade-off introduced by the proposed mechanism.

**Weaknesses:**

1. The paper lacks a detailed description of the defense and attack models, which is essential for readers to fully understand the assumptions and setup of the considered DP-FL system.

2. The proposed RR-Cluster method appears to rely on the assumption that the server is fully honest. However, its effectiveness may significantly degrade—or even vanish—under an honest-but-curious server model, limiting its practical applicability.

3. The final convergence bound presented in Corollary 1 seems overly loose, as it involves multiple assumed constants, reducing its interpretability and practical value.

4. The RR-Cluster method inherently introduces a bias–variance trade-off, which necessitates careful hyperparameter tuning during implementation to achieve an appropriate balance between privacy preservation and model utility.

**Questions:**

1. How can RR-Cluster be adapted to scenarios where the server is honest-but-curious?

2. How does the client sampling rate affect the performance of RR-Cluster? For example, if the sampling rate increases to 0.5, each cluster is likely to become more balanced naturally. In such a case, would RR-Cluster remain effective, or would its advantages diminish, effectively reducing it to a standard federated clustering method without the need for additional rebalancing?

---

> ### Author Response · Authors · 2025-11-21
>
> We thank the reviewer for insightful comments and positive feedback. We hope our response can address your questions.
>
> **[W1: Details of attack/defense model]** We described the threat model and privacy guarantees in Section 2 “Threat Model and Privacy Setting” paragraph. To avoid any ambiguity, in the revision we now restate the setting below: Our setting is the standard and widely-used client-level global DP, as used in DP-FedAvg and most DP-FL work. The adversary we consider is an external observer who sees the final released cluster models and aims to infer the information of participated clients. As required by the reviewer, we revised the “Threat Model and Privacy Setting” paragraph in the main paper, and we highlighted the edits in blue.
>
>
> **[W2: The privacy setting considers a trusted server]** We want to clarify that the used “global DP with a trusted server” is actually the dominant threat model in numerous popular DP-FL works [1, 2, 3, 4]. It is also the practical setting for real deployments in federated learning, where the server operates within some organizations or entities with access control (e.g., hospitals, banks, or service providers). Our method directly targets this scenario, and our evaluation shows that it significantly improves the privacy-utility tradeoff exactly where this model applies.
>
>
> **[W3: Corollary 2 in Appendix]** We would like to clarify that the goal of the analysis is not to give the numerically tightest bound but to make the key tradeoffs explicit: enforcing a minimum cluster size trades off bias (from reassignments) with variance (from DP noise), which helps to explain why RR-Cluster improves utility in practice. We empirically validate the tradeoff across real-world datasets (Shakespeare, CIFAR10. CIFAR100, TinyImageNet, EMNIST, FashionMNIST), where our method consistently outperforms the baselines.
>
>
> **[W4: Needs careful hyperparameter tuning]** We only introduce one additional hyperparameter in this work, and analyze its effects both theoretically and empirically. RR-Cluster is highly robust to the choice of B. In Section 5.3 “Hyperparameter Analysis”, we conduct controlled experiments showing that a very wide range of B values all lead to consistent improvements over the baselines, and the performance curves remain flat around a wide range of B choices.
>
> **[Q1: Adapting RR-Cluster to an honest-but-curious server]** RR-Cluster can be modified to an honest-but-curious server by shifting the noise injection to the clients. Specifically, clients add DP noise to their updates before transmitting to the server. This is mathematically equivalent to first aggregating when adding noise, due to the additive nature of Gaussian noise. The server then performs clustering and rebalancing on these noisy updates. Since the server only observes noisy data, client privacy is preserved against server inspection.
>
> **[Q2: Impact of client sampling rate]** Thanks for the question. Low sampling rates (e.g., q≤0.1) are standard in client level DP-FL to leverage privacy amplification by subsampling [1,5]. Increasing the rate to 0.5 would lead to greater effective noise to maintain the same privacy budget, which may degrade utility significantly. This is the limitation of differential privacy itself, not specific to our method. However, we expect our method to continue to be effective as long as the model updates are not balanced across clusters.
>
> **[References]**
>
> [1] McMahan et al., Learning Differentially Private Recurrent Language Models, In ICLR.
>
> [2] Ceyer et al., Differentially Private Federated Learning: A Client Level Perspective, In NeurIPS.
>
> [3] Wei et al., Personalized Federated Learning with Differential Privacy and Convergence Guarantee, in IEEE TIFS.
>
> [4] Abadi et al., Deep Learning with Differential Privacy, in CCS.
>
> [5] Wang et al., Subsampled Renyi Differential Privacy and Analytical Moments Accountant, In PMLR.

---

### Official Review · Reviewer_YgsE · 2025-11-01

**Soundness:** 3
**Presentation:** 3
**Contribution:** 2
**Rating:** 6
**Confidence:** 3

**Summary:**

The paper introduces RR-Cluster, a lightweight technique for improving privacy-utility tradeoffs in differentially private federated clustering. The core idea is rebalancing the clusters in clustered FL to avoid having small clusters that require adding a lot of noise.

**Strengths:**

- The random rebalancing approach can be integrated with various clustered FL algorithms
- The derivations seem correct and consistent with standard DP theory.
- Consistently outperforms baselines across datasets and privacy budgets.

**Weaknesses:**

- The method assumes that rebalancing doesn't significantly hurt utility, but under concept shift (could be adversarial or not), incorrect assignments could accumulate bias.
- Experiments focus on classification tasks with synthetic and benchmark datasets (despite claim in abstract of the use of real world datasets)
- Only average results reported.

**Questions:**

.-  How would RR-Cluster perform or be adapted in a setting with an untrusted server?
- Can the authors report results for each cluster at least on some experiments? Average results mask how well the model performs.

---

> ### Author Response · Authors · 2025-11-21
>
> We thank the reviewer for the positive and constructive feedback. We hope our response can address your concerns.
>
> **[W1: Concept shift and potential bias from rebalancing]** We agree that concept shift is a critical case, and we already evaluate RR-Cluster exactly in such a setting via the Shakespeare dataset. In our setup, each speaking role defines a client, and different roles use distinct vocabulary even in similar contexts, so the conditional distribution over next characters/words differs across roles. This means that p(y|x) varies across clients, which is a natural concept shifted scenario. We also discussed potential bias and analyzed the bias/variance tradeoff theoretically (the paragraph before Section 4 and Section 4.2) in our paper.
>
>
> **[W2: Real-world datasets]** We clarify that, in addition to standard image benchmarks (FashionMNIST, EMNIST, CIFAR10/100, TinyImageNet), Shakespeare serves as another real-world dataset with natural data partitions across clients: it is derived from actual speaking roles in the plays of William Shakespare, with clients corresponding to real characters rather.
>
> **[W3: Additional results beyond avg accuracy]** Thanks for the suggestion. We extend the main results (Table 1, balanced setting) by further reporting min/max client accuracy across all clients. We put the results in Table 5 of the main paper’s appendix and below:
>
> | Methods | Avg | Max | Min | Avg | Max | Min | Avg | Max | Min |
> |--------|-----|-----|-----|-----|-----|-----|-----|-----|-----|
> |        | **ε = 2** | | | **ε = 4** | | | **ε = 8** | | |
> | DP-FedAvg | 60.88 | 69.20 | 41.94 | 61.50 | 69.38 | 45.71 | 62.72 | 71.14 | 48.65 |
> | DP-IFCA | 62.68 | 75.85 | 08.82 | 64.46 | 76.73 | 21.00 | 65.35 | 79.55 | 22.12 |
> | DP-FeSEM | 63.16 | 74.40 | 16.12 | 64.68 | 77.76 | 18.42 | 64.76 | 78.44 | 24.32 |
> | DP-FedCAM | 49.70 | 57.74 | 18.33 | 56.86 | 66.48 | 21.43 | 61.54 | 72.32 | 29.44 |
> | **RR-Cluster (IFCA)** | **65.69** | 82.86 | 28.42 | **66.63** | 81.12 | 37.83 | **67.51** | 84.38 | 43.24 |
>
> Compared with other clustering based methods, RR-Cluster consistently improves both average accuracy and the min client accuracy, which is the most sensitive metric under DP. Also, the maximum accuracies also remain high, indicating that random rebalancing **does not harm well-aligned clients**.
>
> **[Q1: Adapting RR-Cluster to an untrusted-server setting]** RR-Cluster can be adapted to an untrusted-server scenario by shifting all privacy-critical steps to the client's side locally. In detail, each client would locally clip its model update and cluster-indicator vector, and add DP noise before transmission. The server would then perform clustering with random rebalancing on top of the noisy information, following the post-processing properties of DP and thus privacy-preserving under local DP.
>
> **[Q2: Additional results beyond avg accuracy]**
> Please see our response to W3 for the full min/avg/max client accuracy results. We have added the results in Table 5 of the revised paper.

---

### Official Review · Reviewer_i6N3 · 2025-11-03

**Soundness:** 3
**Presentation:** 2
**Contribution:** 3
**Rating:** 2
**Confidence:** 5

**Summary:**

This paper proposes an add-on technique for existing clustered FL algorithms to enhance their performance when combined with DP. The technique’s core idea is guaranteeing a minimum number of clients assigned to each cluster in each round by random client assignment to smaller clusters. This idea is claimed to reduce the DP noise in the aggregated model updates on server within each cluster. However, this add-on technique induces a trade-off between DP noise reduction and clustering bias. The authors claim that this trade-off does not affect the clustering accuracy considerably.

**Strengths:**

1. The proposed work addresses improving the performance of existing clustered FL algorithms when they are enhanced with DP guarantees, which is an important problem.

2. An extensive set of experimental results are reported (however they need to be improved, see below)

**Weaknesses:**

While I have understood the point of the proposed idea completely, I strongly feel that the current experimental results do not evaluate it properly to validate the correctness of the claims in the paper. I list the existing weaknesses followed by my detailed questions in the next section for clarification.

1. The privacy setting considers a trusted server, which may not always be available in FL settings.

2. The experimental results are reported in an optimistic way that does not fully evaluate the proposed idea. For example, average accuracy is not enough and can be extended with worst accuracy across clients to investigate the effect of the proposed idea. Also, the used baselines can be extended/replaced with SoTA ones.

3. The theoretical results about privacy guarantees do not have a clear and important message, and it mainly uses the composition property of RDP.

4. The convergence analysis relies on strong convexity of loss functions. Yet, it has a clearer message than the privacy analysis.

5. The chosen baselines can be improved. Current baselines (some SoTA are missing), metrics (avg accuracy) and experimental results (simple datasets) do not fully evaluate the proposed idea. Based on my experience, MNIST. FMNIST, EMNIST are easy dataset for clustered FL, compared to colured datasets like CIFAR10, Tiny ImageNet, ….

I explain further in my questions below.

**Questions:**

1. The main point in the proposed idea is that, considering the global DP setting and the trusted server, ensuring a minimum size for clients clusters in each round reduces the DP noise added by the server to the aggregated parameter within each cluster. As the authors claim, this improves performance. However, improving the performance of existing methods is valuable if it enables the methods to beat the existing SoTA approaches. Following this point, three main questions rise:

1.1. The current data heterogeneity are “suitable” for the proposed random rebalancing idea. If the level of data heterogeneity across clusters is high, randomly mixing clients from different ground-truth clusters could affect utility severely. For example when concept shift exists across clusters (i.e. when $p(y|x)$ varies, see [1]). This is usually the assumption in clustered FL settings, otherwise personalized FL techniques could be used instead of clustered FL techniques (explained in the second question below) for moderate data heterogeneities. So investigating such data distributions will fully evaluate the applicability of the idea.

1.2. Even if the data heterogeneity across clusters is not high (like the rotation and natural heterogeneities considered in the current results) and if aggregation for a larger number of clients reduces the effective DP noise, why not to use the MR-MTL (its DP extension [1,2], which is also shown in [1] to largely improve DP-FedAvg for sample-level DP)? This personalization technique is stronger than the considered FedPer, and has both the DP noise reduction (by aggregation across “all” clients on the server) as well as personalization (without even introducing “three” hyperparameters. Remember, RR-cluster introduces three hyper parameters $c_s$, $c_{\theta}$ and $B$). Including this baseline is important because it benefits from the same noise reduction mechanism that RR-cluster does.

1.3. The current used metric is the average accuracy. Reporting the best, average and worst test accuracy across clients will make the effect of incorporating random balancing clearer.

[1] S. Malekmohammadi, et. al., Differentially Private Clustered Federated Learning, TMLR 2025.

[2] Z. Liu, et. al., On Privacy and Personalization in Cross-Silo Federated Learning, Neurips 2022.

2. The idea proposed in the work [1] above is for sample-level DP with untrusted server, yet it can directly be applied to truster server and global DP settings too, i.e. using a full batch size in the first round by clients followed by small batch sizes in the next rounds. This especially helps because in the considered trusted-server setting, clients send their “clean” model updates with no DP noise to the server, so using the full batch size idea in [1] can make detection of underlying clusters easy for the server. This would also be a strong baseline to consider as a SoTA.

3. In line 188, it is mentioned that “Note that we can set $B$ so that any client update can only be sampled and reassigned to another cluster at most once”. Could the authors clarify further about this? Also, how is the client sampling from large clusters done? i.e. uniform without replacement? What if all are sampled from one “large” cluster, making it “small”?

4. Line 377 implies that hyper parameters (at least the important ones like learning rate) are not set carefully for all algorithms. At least for the most strong baselines (recommended above) this should be the case. Also, in line 363, where is the set $k \in \\{2,4\\}$ is coming from? In lines 383 to 365, you consider $k \in \\{3,4\\}$ clusters, which is close to the above set chosen for experiments. Remember that the recommended MR-MTL baseline does not need to know number of clusters or an approximate of that. Also, the work [1] can approximate the underlying number of clusters, and it can do this even better especially in the considered trusted server setting that clients send their “clean” model updates to the server with no DP noise.

5. In tables 1 (and 6), why do we observe improvement even when clusters are “balanced”? In this case, random rebalancing makes some clusters smaller and some larger (non-uniform).
minor comments:

typo in line 036: clustered models -> cluster models
line 269: theorem 1 -> proposition 1
line 1011: Table 10 -> Table 5

---

> ### Author Response · Authors · 2025-11-21
> **Part 1 of our Response (W1-W4)**
>
> Thanks for the feedback and questions. We respond to each of the reviewer’s concerns as follows.
>
> **[W1: The privacy setting considers a trusted server]** While we do not expect our setting cannot cover all possible FL applications, we want to clarify that this “global DP with a trusted server” model is the dominant threat model in numerous popular DP-FL works [e.g., 1, 2, 3, 4]. We also described this notion and its implications throughout the paper; see, e.g., Definition 1 and the “Threat Model and Privacy Setting” paragraph before Section 3. We note that in real world scenarios, an external observer can always pose a threat (e.g. infer training participants), and the standard client-level global DP directly addresses this threat by guaranteeing that the final cluster models are insensitive to any single client.
>
> **[W2: Add Min/Max client accuracy]** We extend the main results (Table 1, balanced setting) by further reporting the min/max client accuracies across all clients. We organize the additional results in Table 5 in the appendix of our revision, as well as in the table below:
>
> | Methods | Avg | Max | Min | Avg | Max | Min | Avg | Max | Min |
> |--------|-----|-----|-----|-----|-----|-----|-----|-----|-----|
> |        | **ε = 2** | | | **ε = 4** | | | **ε = 8** | | |
> | DP-FedAvg | 60.88 | 69.20 | 41.94 | 61.50 | 69.38 | 45.71 | 62.72 | 71.14 | 48.65 |
> | DP-IFCA | 62.68 | 75.85 | 08.82 | 64.46 | 76.73 | 21.00 | 65.35 | 79.55 | 22.12 |
> | DP-FeSEM | 63.16 | 74.40 | 16.12 | 64.68 | 77.76 | 18.42 | 64.76 | 78.44 | 24.32 |
> | DP-FedCAM | 49.70 | 57.74 | 18.33 | 56.86 | 66.48 | 21.43 | 61.54 | 72.32 | 29.44 |
> | **RR-Cluster (IFCA)** | **65.69** | 82.86 | 28.42 | **66.63** | 81.12 | 37.83 | **67.51** | 84.38 | 43.24 |
>
> Compared with other clustering based methods, RR-Cluster consistently improves both average accuracy and the min client accuracy. Also, the maximum accuracies remain high, indicating that random rebalancing does not harm the clients who benefit from clustering.
>
> **[W3: On the privacy analysis]** Considering that our method adds a new component (the rebalancing operation) that is not present in existing DP-FL methods, we provide the privacy analysis to explicitly show that 1) each step of RR-Cluster complies to the privacy setting, and 2) how the privacy budget accumulates over rounds. For private clustering works, the output is a set of k clustering models, which is different from the classic setting of outputting one model where privacy analysis is straightforward. We therefore think it is critical for readers to understand our method as well as privacy guarantees.
>
> **[W4: Strong convexity assumption]** We first want to note that the strong convexity assumption is a standard and common practice in convergence analyses of federated optimization [5], differentially private federated learning [3], and federated clustering algorithms [6]. In the clustering scenario, strong convexity is especially necessary, for instance, our current proof needs this strong-convexity assumption to guarantee a sufficient difference between the loss values of different clusters (see Equations 12-16, Appendix B.1). We would like to note that we can generate useful insights from our analysis that imply the tradeoffs between bias and variance due to the additional parameter B; please see discussions at the end of Section 4 for details.
>
> Additionally, while our theory focuses on the convex case, RR-Cluster shows superior empirical performance in non-convex experiments. This suggests that our core intuition that rebalancing reduces DP noise at the cost of potential bias still holds in practical scenarios.

---

> > ### Author Response · Authors · 2025-11-21
> > **Part 2 of our Response (W5, Q1.1)**
> >
> > **[W5: Additional datasets]** First, we would like to clarify that colored image experiments (on CIFAR10 and CIFAR100) suggested by the reviewer are already included in our initial submission experiments (see Section 5.1 of the main text, and Table 5 of the Appendix E.1). These results show that RR-Cluster consistently outperforms strong DP baselines on standard colored-image benchmarks. In addition to image datasets, we also included text data and a wide coverage of underlying data partition/clustering structures in the original submission (Section 5).
> >
> > In response to the reviewer’s specific suggestion about TinyImageNet, we have added new experiments on this dataset with 1000 clients and 4-direction image rotation setting on a CNN (same as other image datasets) and the results are in the table below.
> >
> > | Methods | CIFAR10 (ε=4) | CIFAR10 (ε=8) | CIFAR100 (ε=4) | CIFAR100 (ε=8) | TinyImageNet (ε=4) | TinyImageNet (ε=8) |
> > |---------|---------------|---------------|-----------------|----------------|---------------------|---------------------|
> > | DP-FedAvg | 52.89 | 54.48 | 14.05 | 14.49 | 13.67 | 14.06 |
> > | DP-FedProx | 52.04 | 54.61 | 14.48 | 14.53 | 14.01 | 14.12 |
> > | DP-IFCA | 53.91 | 56.77 | 16.29 | 17.47 | 15.79 | 16.76 |
> > | DP-FeSEM | 53.01 | 55.87 | 16.23 | 17.08 | 15.65 | 16.44 |
> > | DP-FedCAM | 52.50 | 54.73 | 16.36 | 17.04 | 15.71 | 16.38 |
> > | **RR-Cluster (IFCA)** | **57.46** | **58.41** | **17.83** | **18.00** | **17.14** | **17.90** |
> >
> > We see that RR-Cluster remains the highest accuracy under all privacy budgets, showing that random rebalancing is also effective in more difficult tasks. We have integrated all CIFAR10, CIFAR100, and TinyImageNet results into Table 6 of the revised paper.
> >
> >
> > **[Q1.1 On the level of data heterogeneity]** We would like to clarify that we did not create heterogeneity in the way that favors our method. Instead, we evaluate RR-Cluster under multiple types and levels of heterogeneity following either natural partitions or setups in previous works, including exactly the concept shift (with a changing p(y|x)) setting the reviewer refers to. We elaborate on these settings as follows.
> >
> > **(1) Table 2: Commonly-used heterogeneous settings** Rotation-based heterogeneity is standard in federated clustering literature [6, 7, 8], and our results show consistent improvements across these widely-adopted non-IID configurations.
> >
> > **(2) Table 4: Systematic control of heterogeneity strength, including medium and high heterogeneity** Table 4 varies label skewness, covering mild, medium, and high heterogeneity. Furthermore, the right side of Table 4 increases heterogeneity by varying cluster imbalance. In all these scenarios, RR-Cluster continues to outperform existing DP clustering baselines, including under “high heterogeneity,” which directly contradicts the concern that our method only works under mild non-IIDness.
> >
> > **(3) Table 3: Shakespeare dataset, which contains concept shift (different P(y|x))** Shakespeare is a next-character prediction task where different roles use different vocabularies and linguistic styles. This corresponds to the reviewer’s suggesting of evaluating in settings with concept shift, i.e., different P(y|x).
> >
> > In summary, our evaluation in the original submission already includes: 1) standard data heterogeneity settings (Table 2), 2) systematically increasing heterogeneity with label-skew (Table 4), and 3) concept-shifted distributions with differing P(y|x) (Table 3). Across all these heterogeneous regimes, RR-Cluster consistently improves DP clustering performance.

---

> > > ### Author Response · Authors · 2025-11-21
> > > **Part 3 of our Response (Q1.2-Q3)**
> > >
> > > **[Q1.2: MR-MTL Baseline]** We appreciate the suggestion of adding the MR-MTL baseline, but we note that the MR-MTL method is not comparable with ours, as it is focusing on sample-level, as opposed client-level DP. We explain in detail below
> > > **1) [Privacy definition] Different Privacy Units for Different Privacy** Our work explores client-level global DP, where neighboring datasets differ by adding or removing an entire client, and the outputs are a small set of cluster models. In contrast, MR-MTL is designed for sample-level DP in cross-silo FL, implemented via running DP-SGD on per-example gradients locally. As emphasized in standard DP tutorial [9], once the “privacy unit” (record vs. user) changes, the implications of $\epsilon$’s also change, hence accuracy comparisons “under the same $\epsilon$” across these two notions are meaningless.
> > >
> > > **2) [Implementation and Output Space] Its privacy guarantees cannot be applied to global DP setting (so does the noise scale & composition)** MR-MTL’s DP guarantee is local to each silo’s training examples, and it outputs a model per client, as opposed to k cluster models for n clients where k << n.
> > > Our definition is global DP on the final released k cluster models, protecting client participation. Switching the DP unit from “sample” to “client” and changing the output space from n models to k models requires different sensitivity bounds, noise scales, and composition rules.
> > >
> > > For these reasons, MR-MTL cannot be placed into our client-level DP setting without redesigning its entire DP mechanism and objective.
> > >
> > > **[Q1.3: Add Min/Max client accuracy]**
> > >
> > > Please see W2 for the additional metrics.
> > >
> > > **[Q2: DPCFL Baseline]** Thanks for suggesting this related work. We have cited it in our revision. Similar to MR-MTL, DPCFL is not in the same privacy setting as our work and cannot be directly used as a baseline in the client-level global DP framework.
> > >
> > > **1) [Theoretically] Different Privacy Units for Different Privacy** DPCFL is designed for sample-level DP, where each client runs DP-SGD locally and all information sent to the server (model updates and cluster selections) satisfies DP w.r.t. a single record in that client’s dataset. Consequently, the privacy mechanism in DPCFL and in our setting corresponds to fundamentally different guarantees, which are not transferrable and comparable.
> > >
> > > **2) [Implementation] Mechanism and noise calibration are incompatible with global client-level DP** The “full batch in the first round + small batches afterwards” idea in DPCFL is interesting, but it is specifically analyzed under sample-level DP-SGD noise to help the server cluster clients’ noisy model updates via GMM in the first round. Comparing with DPCFL would require re-defining adjacency, re-deriving sensitivity and noise scales, and redoing the privacy accounting for all steps under global DP—this would effectively be a new algorithm rather than the original DPCFL.
> > >
> > > Given that DPCFL is a complementary sample-level DP clustered FL method, we do not compare our approach with it.
> > >
> > > **[Q3: Details of the Random Rebalancing]**
> > >
> > > Thanks for the questions. We provide additional clarification here as follows.
> > >
> > > **(1) Why each client can be sampled at most once** To control the sensitivity of aggregated model updates within each cluster, we need to make sure that each client update can affect at most two clusters per round. Sampling any client multiple times would break the sensitivity bound. Therefore, we explicitly design the sampling rule (as detailed below) to guarantee that any client can only be reassigned at most once at each iteration.
> > >
> > > **(2) How sampling is performed** For each large cluster​, we partition its clients uniformly random into: 1) an unsampleable set of size B (guaranteeing the cluster never has clients less than B), 2) sampleable set, from which we may reassign clients to small clusters. We then uniformly sample without replacement from the sampleable set only. This ensures randomness for every client while also maintaining a hard lower bound of B per cluster.
> > >
> > > **(3) A large cluster will not be smaller than B after resampling** We use the unsampleable set to enforce at least B clients for all large clusters, so even in the worst-case sampling outcome, no large cluster can be reduced below the minimum threshold B.

---

> > > > ### Author Response · Authors · 2025-11-21
> > > > **Part 4 of our Response (Q4, Q5)**
> > > >
> > > > **[Q4: On the choice of number of clusters]** We thank the reviewer for pointing out potential confusion around the choice of cluster numbers. We clarify below that (1) the search space of k ($\in {2,3,4}$) is a standard search space in the federated clustering literature [6,7], especially in such rotation-based tasks. Because our method is designed as an add-on to these algorithms, we follow the same hyperparameters as the base algorithms for a fair comparison. (2) All other experiments (Dirichlet Heterogeneity in Table 3, and Natural Heterogeneity with Concept Shift in Table 2) do not contain any explicit cluster structure, and thus RR-Cluster does not use any additional information about the “true” number of clusters that the baselines do not have.
> > > >
> > > > **[Q5: Improvements when clusters are balanced]** The “balanced” setting in Table 1 refers to the ground-truth balanced cluster distribution, not the per-round cluster sizes produced by the clustering algorithm. Due to DP noise and the well-known cluster-collapse issue, some clusters can be under-trained (thus having very small cardinalities during training) even when the underlying data are balanced across clusters. RR-Cluster brings the effective cluster sizes closer (by resampling) to the ground truth and preventing collapse. We further discussed this point in detail in Appendix E.5 and Table 11 of our initial submission, and show that our resampling stabilizes training, which explains why RR-Cluster could improve performance in both balanced and unbalanced cases.
> > > >
> > > > **[Minor: Typos]** We thank the reviewer for pointing out these minor issues. We have corrected all mentioned typos in the revised manuscript and highlighted the changes in blue for clarity.
> > > >
> > > > **[References]**
> > > >
> > > > [1] McMahan et al., Learning Differentially Private Recurrent Language Models, In ICLR.
> > > >
> > > > [2] Ceyer et al., Differentially Private Federated Learning: A Client Level Perspective, In NeurIPS.
> > > >
> > > > [3] Wei et al., Personalized Federated Learning with Differential Privacy and Convergence Guarantee, in IEEE TIFS.
> > > >
> > > > [4] Abadi et al., Deep Learning with Differential Privacy, in CCS.
> > > >
> > > > [5] Li et al., On the Convergence of FedAvg on Non-iid Data, in ICLR.
> > > >
> > > > [6] Ghosh et al., An Efficient Framework for Clustered Federated Learning, in NeurIPS.
> > > >
> > > > [7] Ma et al., Structured Federated Learning through Clustered Additive Modeling, In NeurIPS.
> > > >
> > > > [8] Long et al., Multi-center Federated Learning: Clients Clustering for Better Personalization, In WWW.
> > > >
> > > > [9] Cynthia et al., The Algorithmic Foundations of Diﬀerential Privacy, In FTTCS.

---

### Author Response · Authors · 2025-12-03
**General Response to AC**

We thank the AC for meta-reviewing our work. Across all four reviewers, they agree on several strengths of the paper: (i) our target problem: improving clustered FL under DP is important; (ii) the proposed RR-Cluster is a simple, plug-and-play mechanism that can be integrated into multiple clustered FL algorithms; (iii) our theoretical analysis is consistent with standard DP theory and explicitly captures the bias–variance tradeoff introduced by rebalancing; (iv) and our empirical results consistently show improved accuracy across datasets and privacy budgets. We would like to summarize our rebuttal process below.

Reviewer i6N3 suggested adding more datasets, more metrics, and comparisons with two baselines (MR-MTL and DPFCL). In the revision, despite we already had 5 datasets across both text and image domain in the original submission, we added TinyImageNet experiments in Table 6 and showed consistent improvements. We also added additional metrics in Table 5, showing that our RR-Cluster improves not only the average but also the worst-client accuracy. We have cited the suggested references and clarified that they focus on a completely different sample-level DP setting and thus are not comparable to our standard client-level DP setup.

Reviewer YgsE raised questions about dataset properties and client-level statistics, as well as how RR-Cluster might be extended to a different setting with an untrusted-server. We clarified that our Shakespeare experiments in Table 2 already instantiate a natural concept-shift scenario, added more matrics/statistics, and added discussions around how RR-Cluster can be used in the presence of an honest-but-curious server by adding noise locally.

Reviewer ax89’s main concern was about the tightness of the convergence bound and an additional hyperparameter B. We clarified that the purpose of our bound is to explicitly explore the bias–variance tradeoff introduced by B rather than to provide the tightest asymptotic rate. We also emphasized, supported by a dedicated ablation study in Section 5.3, that RR-Cluster is not overly sensitive to hyperparameters: the method remains robust across a wide range of B values and does not require fine-grained tuning in practice.

Reviewer NBjo raised major questions around the differences between our convergence rate and previous vanilla non-private clustering method’s rate and the interpretation of Theorem 2. In our response, we clarified that our analysis is done in a strictly harder setting than prior works, which introduces additional error terms and therefore cannot recover the non-private rate. We discussed in more detail the interpretation of Theorem 2 and Corollary 1. We also cleaned up the notation and fixed typos.

Overall, we feel the reviewers did not raise major criticisms of our work, and we believe the revision addresses all comments raised by the reviewers, by adding more datasets, reporting more metrics, and clarifying the questions. We hope our summary is useful for your meta-reviewing.

Appreciatively,
Authors

---

### Meta-Review · Area_Chair_FuK2 · 2025-12-23

**Summary:**

The paper proposes aa nalgorithm for Federated Clustering that enforces minimum cluster sizes via random rebalancing to lower the sensitivity-to-noise ratio in Global DP settings.

Strengths:
- The proposed method is easy to implement on top of existing clustering algorithms.
- Reviewers generally acknowledged that the method outperforms baselines
Weaknesses:
- Reviewer NBjo noted that the convergence bound (Theorem 2) explodes as $B \to 0$ (due to $1/B$ terms), whereas the algorithm itself reduces to the base DP-IFCA method at $B=0$. This issue has only been partially addressed in the rebuttal and would require a major revision
- Reviewers pointed out some limitation in the experimental setting that have been only partially addressed in the rebuttal.

Overall, the paper is interesting but it is not yet ready for publication in the current form.

**Reviewer Concerns:**

Additional experiments have been run but they do not fully respond to the critiques of reviewer i6N3 and reviewer YgsE.

Concerns about Theorem 2 are not fully addressed.

**Reviewer Scores:**

I think that reviewer i6N3 would have not change their score as their comments have been only partially addressed.

Similarly I would not have expected reviewer YgsE to change their score.

I also feel the other two reviewers probably would not have raised their score or only marginally.

---

### Decision · Program_Chairs · 2026-01-26

Reject